# Effects of SGLT-2 inhibitors on renin-angiotensin-aldosterone system and their correlation with glucose metabolism in type 2 diabetes mellitus patients with hypertension: A prospective study

Ningning Wang[1][ID][⚭][*], Junfeng Kong[1][⚭], Ziming Lu[1], Fei Cao[2], Erjun Tian[1], Junhui Li[1], Shutong Li[1], Shuai Liu[3,4,5], Baohong Yue[3,4,5][*]

**1** Department of Laboratory Medicine, the First People's Hospital of Pingdingshan, Pingdingshan, Henan, China, **2** Department of Orthopedics, the First People's Hospital of Pingdingshan, Pingdingshan, Henan, China, **3** Department of Laboratory Medicine, the First Affiliated Hospital of Zhengzhou University, Zhengzhou, Henan, China, **4** Faculty of Laboratory Medicine, the First Clinical Medical College, Zhengzhou University, Zhengzhou, Henan, China, **5** Key Clinical Laboratory Medicine of Henan Province, Zhengzhou, Henan, China

⚭ NW and JK are contributed equally to this work.
* fccybh@zzu.edu.cn (BY); hourinsunninng@163.com (NW)

## Abstract

### Background

The impact of sodium-glucose cotransporter-2 inhibitors (SGLT-2is), including dapagliflozin, on the renin-angiotensin-aldosterone system (RAAS) in type 2 diabetes mellitus (T2DM) patients remains controversial, as they may either activate or inhibit RAAS, subsequently influencing glucose metabolism and the accuracy of the aldosterone-to-renin ratio (ARR) in diagnosing primary aldosteronism (PA). However, the effects of SGLT-2is therapy on RAAS and their correlation with glucose metabolism have not been well studied.

### Methods

A cohort of 147 patients with T2DM and hypertension was prospectively recruited and categorized into two groups: diabetic kidney disease group (DKD, n = 73) and non-DKD group (n = 74), based on diagnostic criteria for diabetic complications. Patients were prescribed 10 mg dapagliflozin daily for 3 months. The primary outcome measure was the change in renin (REN) levels during outpatient visits at baseline, 1 month, and 3 months. The secondary outcome was the change in other metabolic biomarkers from baseline to the 3-month visit. To evaluate the relationship between RAAS components and various glucose metabolism indicators, including HbA1c,

**Data availability statement:** All the data are available at https://github.com/ningningwang688/SGLT-2is-on-T2DM.

**Funding:** This study was supported by the National Natural Science Foundation of China (grant numbers 82474325). Author FC is a participant in the grant program and contributed to project administration and manuscript review. The other authors have no relevant competing interests. The funders had no role in study design, data collection and analysis, decision to publish, or preparation of the manuscript.

**Competing interests:** The authors have declared that no competing interests exist.

FBG, CP, HOMA-β, HOMA-IR, and UACR, Spearman correlation and multiple linear regression analyses were conducted at baseline and 3-month visit.

## Results

After 3 months, the BMI, HbA1c, FBG, TG, TCHO, SBP, DBP, UACR, and HOMA-IR levels were significantly decreased, while Crea and HOMA-β were significantly increased from the baseline in both groups. Additionally, the differences from the baseline in FBG ($-2.64 \pm 2.66$ vs. $-1.70 \pm 1.92$ mmol/L) and UACR ($-355.01 \pm 1534.12$ vs. $-4.66 \pm 7.86$ mg/g) values were significantly higher in DKD group than those in non-DKD group. REN levels increased significantly from baseline at 1-month visit ($4.15 \pm 7.35$ vs. $2.75 \pm 8.03$ ng/L; DKD vs. non-DKD; between-group difference, $p < 0.05$), while ARR values decreased significantly from baseline at 1-month visit ($-0.70 \pm 1.16$ vs. $-0.59 \pm 1.19$; between-group difference, $p > 0.05$) and 3-month visit ($-0.45 \pm 1.15$ vs. $-0.42 \pm 1.07$; between-group difference, $p > 0.05$) in both groups (within-group change vs. baseline, $p < 0.017$). Interestingly, no temporal differences were observed in ALD levels. REN levels returned to baseline after three months of treatment; yet the ARR, which was primarily influenced by REN, remained below its initial value. Multiple linear regression analysis revealed that a 100% increase in Log-ALD was associated with a 0.143 nmol/L higher CP and a 15.8% higher HOMA-IR in DKD group($p < 0.05$). A 100% increase in Log-REN was associated with a 0.359 mg/g lower UACR in DKD group and a 0.042 mmol/L higher FBG in non-DKD group($p < 0.01$). These correlations were independent of the internal interactions of RAAS but were significantly attenuated after 3 months of treatment.

## Conclusions

The correlations between RAAS markers and glucose metabolism indices were significantly attenuated, potentially due to lowered blood glucose levels after SGLT-2is treatment. However, it is noteworthy that short-term therapy may elevate REN levels and reduce ARR, potentially resulting in false-negative outcomes in PA screening.

## 1. Introduction

Type 2 diabetes mellitus (T2DM) represents the most common metabolic disorder [1], characterized by chronic hyperglycemia that arises from insufficient insulin secretion or insulin resistance [2]. Diabetic kidney disease (DKD), a major microvascular complication, is rising in incidence annually and is the leading cause of mortality in this population [3]. Additionally, primary aldosteronism (PA) occurs in 11.3–14% of T2DM patients with hypertension [4,5], and is associated with a higher prevalence of diabetes compared to essential hypertension [6]. Both PA and T2DM can cause significant damage to vital organs, including the heart, brain, and kidneys, thereby increasing the risk of cardiovascular and cerebrovascular events [6]. Aldosterone-to-renin ratio (ARR) is popularly used for screening PA but is influenced by various factors,

including age, potassium levels, body posture, and certain medications [7,8]. Current diagnostic guidelines for PA recommend discontinuing antihypertensives that affect the renin-angiotensin-aldosterone system (RAAS) for 2–4 weeks before assessing ARR or switching to antihypertensive agents, such as α-blockers, with less RAAS effect [7,8].

Sodium-glucose cotransporter-2 inhibitors (SGLT-2is) are oral antidiabetic medications that lower blood glucose levels by inhibiting the reabsorption of glucose and sodium in the proximal convoluted tubule, thereby increasing glycosuria [9]. SGLT-2is effectively improve hyperglycemia without directly acting on β-cells, potentially slowing β-cell failure by decreasing the excessive glucose influx into these cells [2]. Furthermore, these inhibitors provide cardio-renal protective effects [9]. By reducing sodium-coupled glucose reabsorption in the proximal convoluted tubule, SGLT-2is enhance the distal delivery of sodium to the macula densa, theoretically leading to decreased renin levels [10] and exerting a RAAS-inhibition-like effect [11], thereby slowing the progression of DKD. The osmotic diuresis and natriuretic effects of SGLT-2is, along with the associated reductions in extracellular fluid volume and blood pressure, may activate RAAS to maintain normal glomerular filtration rates [11]. However, the precise pathophysiological mechanisms of SGLT-2is on the RAAS have not been clearly defined. Clinical evidence regarding the relationship between SGLT-2is and RAAS activation remains controversial and contradictory [12]. A study found no significant effect on aldosterone (ALD), renin (REN) levels, or ARR following 2–6 months of SGLT-2is therapy [13]. Conversely, another study reported increased REN levels without corresponding changes in ALD levels after 198 days of SGLT-2is treatment [10]. It is still uncertain whether SGLT-2is should be discontinued before ARR measurement.

RAAS acts as a paracrine, autocrine, and endocrine system in various tissues and plays a crucial role in regulating blood pressure, glucose metabolism, and the functions of both the kidneys and pancreatic β cells in diabetes [1,11,14,15] (Fig 1). Hyperglycemic elevates the succinate receptor GPR91 expression in the kidney, subsequently activates paracrine signaling pathways, and ultimately leads to the release of renin [16,17]. Griffin TP study, which did not address the use of SGLT-2is, revealed that increased renin levels were positively correlated with higher HbA1c values [18]. Durvasula RV et al. found that high blood glucose levels increased renin and angiotensin II (AII) activity, resulting in sustained hypertension, glomerular hyperfiltration and hyperperfusion, as well as progressive podocyte damage and loss in DKD [19,20]. Preclinical studies demonstrated that ALD impaired β-cell function and insulin secretion by inhibiting insulin signaling pathways and diminishing glucose-stimulated insulin secretion [14,15]. These studies suggest that the relationship between RAAS and glucose metabolism is intricate and bidirectional before SGLT-2is treatment.

Hyperglycemia may activate the RAAS, whereas the activation of RAAS can induce insulin resistance, impair renal function, and aggravate glucose metabolism disorders [1]. SGLT-2is may affect these pathways by reducing blood glucose levels and protecting β-cells and renal function through the induction of glycosuria [1] (Fig 1). However, the interplay between SGLT-2is, RAAS, and glucose metabolism has not yet been reported. This interaction may further complicate the clinical interpretation of ARR test. Previous researches exploring the relationship between RAAS and glucose metabolism have primarily utilized data that either did not involve the use of SGLT-2is or was derived from retrospective studies [18–22]. Consequently, this prospective self-matched study was undertaken to investigate the effects of SGLT-2is on RAAS and their associations with glucose metabolism, aiming to improve the clinical interpretation of ARR test in patients with T2DM and hypertension.

## 2. Methods

### 2.1. Study subjects

Patients diagnosed with T2DM and hypertension were identified and recruited through convenience consecutive sampling at routine diabetes outpatient visits to the First People's Hospital of Pingdingshan between April 2024 and November 2024. After obtaining written informed consent, eligible patients were prescribed dapagliflozin at a daily dosage of 10 mg for a period of three months. The RAAS and other relevant indicators were measured during outpatient visits at baseline, 1 month, and 3 months. The flow diagram of patient recruitment was shown in Fig 2. Ultimately, a total of 147 patients with T2DM were enrolled in the study.

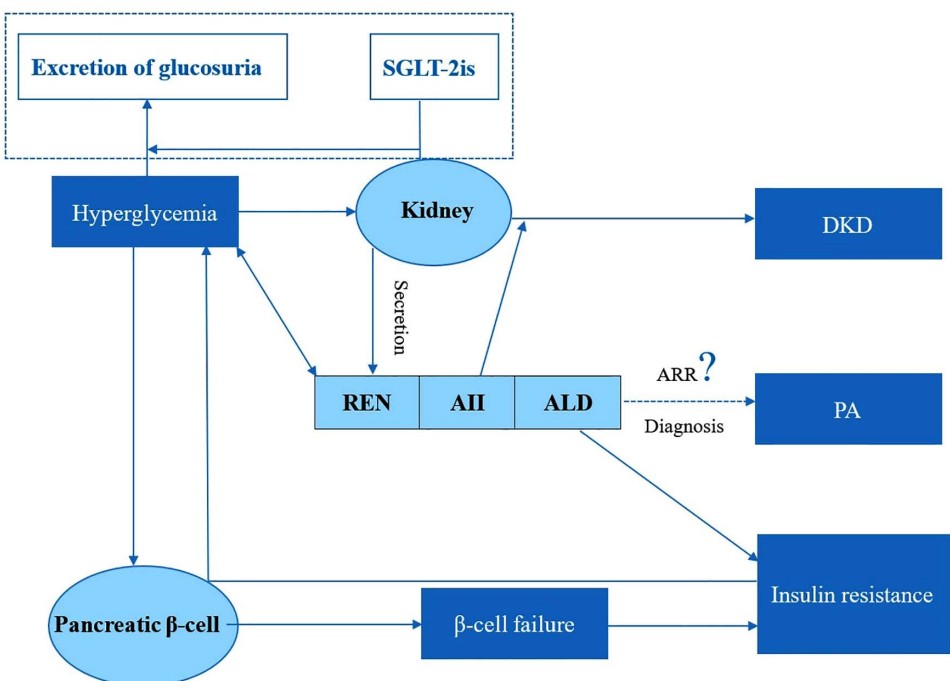

**Fig 1. The complex interplay between SGLT-2is, RAAS, and glucose metabolism in type 2 diabetes mellitus patients.** Hyperglycemia may activate the RAAS, whereas the activation of RAAS can induce insulin resistance, impair renal function, and aggravate glucose metabolism disorders. SGLT-2is may affect these pathways by reducing blood glucose levels and protecting β-cells and renal function through the induction of glycosuria. SGLT-2is, sodium-glucose cotransporter-2 inhibitors; RAAS, renin-angiotensin-aldosterone system; DKD, diabetic kidney disease; PA, primary aldosteronism.

The study protected patient privacy by utilizing a numerical identifier to anonymize personal information, received ethical approval from the Ethics Committee of the First People's Hospital of Pingdingshan (PYLL2024032604), and adhered to the principles of the Declaration of Helsinki. All patients were briefed about the study and provided their written informed consent for participation. Detailed timeline of the study was as follows: Ethical approval of the study: March 26, 2024. Recruitment period of the study: from April 2, 2024, to November 29, 2024.Completion of the study: June 15, 2025.

Based on the diagnostic criteria for diabetic complications [23], the patients were divided into two groups: the diabetic kidney disease group (DKD, n = 73) and the non-DKD group (n = 74). The DKD group comprised 38 males and 35 females, with a mean age of 53.27 years and a median diabetic duration of 81 months. Additionally, the non-DKD group included 40 males and 34 females, with a mean age of 55.76 years and a median diabetic duration of 72 months.

## 2.2. Inclusion criteria and exclusion criteria

Inclusion criteria for the study were as follows: (1) Diagnosis of T2DM according to WHO guidelines [23]; (2) Essential hypertension defined as systolic blood pressure≥140 mmHg and/or diastolic blood pressure≥90 mmHg, or with a previous diagnosis and a history of treatment with antihypertensive medications; (3) Patients aged between 18 and 80 years; (4) Patients had not taken any agents that could markedly affect the ARR, such as angiotensin converting enzyme inhibitors, angiotensin receptor blockers, diuretics, or calcium channel blockers within the last 2 months, or aldosterone receptor antagonists, β-blockers, or direct renin inhibitors within the last 4 months [8]; For patients with poorly controlled blood pressure (≥160/100 mmHg), medications with less effect on RAAS, such as α-blockers (e.g., terazosin), were administered. (5) Normal thyroid function. Exclusion criteria included: (1) Patients with contraindications to dapagliflozin therapy

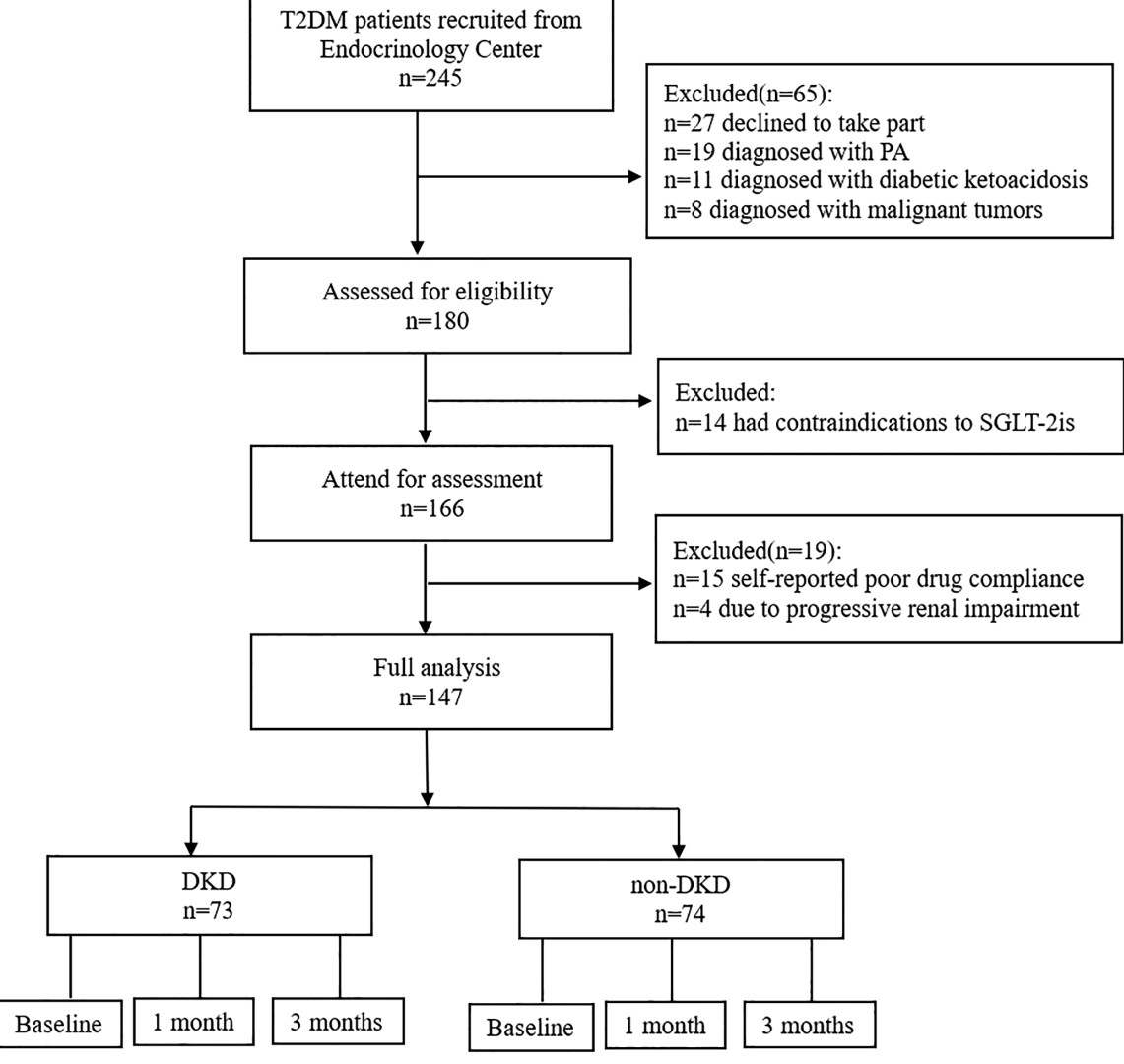

**Fig 2. Flow diagram of patient recruitment.** T2DM, type 2 diabetes mellitus; PA, primary aldosteronism; DKD, diabetic kidney disease.

(experiencing side effects such as drug hypersensitivity, complicated urinary tract infections or genital infections, and acute kidney injury); (2) Patients who discontinued dapagliflozin treatment or added above agents that significantly affect the RAAS during the study period; (3) Patients experiencing acute infections, diabetic ketoacidosis, hyperosmolar hyperglycemic syndrome, or other related complications; (4) Patients with stage IV or V chronic kidney disease, or undergoing renal replacement therapy or transplantation; (5) Patients with primary aldosteronism; (6) Patients with secondary hypertension, congestive heart failure, severe liver dysfunction, or malignant tumors; (7) Patients with Cushing's syndrome or hyperparathyroidism; (8) Pregnant or breastfeeding women.

## 2.3. Method of detection

According to the blood collection criteria for ARR test [8], samples were collected in the morning after patients had been up and ambulatory for at least 2 hours and seated for 5~15 minutes. The plasma concentrations of angiotensin II (AII),

renin (REN), and aldosterone (ALD) were measured by the chemiluminescence method (Zhengzhou AutoLumo A2000 Plus, Autobio, China). Based on the diagnostic guidelines for PA [8], when ALD was measured in ng/dL and REN in ng/L, the common cut-off value for the ARR was approximately 5.7. In our study, both REN and ALD were measured in ng/L. Since 1 ng/L equals 10 ng/dL, the ARR was calculated using the formula: $ARR = ALD/(REN \times 10)$. Fasting blood glucose (FBG), C-peptide (CP), triglycerides (TG), total cholesterol (TCHO), high-density lipoprotein cholesterol (HDL), low-density lipoprotein cholesterol (LDL), urea nitrogen (Urea), creatinine (Crea), potassium (K), sodium (Na), calcium (Ca), phosphorus (P), and other biochemical indicators were quantified utilizing the Roche Cobas 8000 automated analyzer (Roche, Germany). Homeostasis Model Assessment of β-cell function (HOMA-β) and Homeostasis Model Assessment of Insulin Resistance (HOMA-IR) were computed using CP and FBG levels via software available at http://www.dtu.ox.ac.uk/homacalculator/.

The random urinary albumin-to-creatinine ratio (UACR) was assessed utilizing the BA400 automatic specific protein analyzer (Biosystems, Spain). The measurement of HbA1c (Hemoglobin A1c) was performed using the Shanghai Huizhong MQ-6000 analyzer. Systolic blood pressure (SBP), diastolic blood pressure (DBP), height (m), and weight (kg) were measured using standardized techniques, and body mass index (BMI) was calculated accordingly. All tests were processed and analyzed concurrently with the daily specimens according to ISO (International Organisation for Standardisation) 15189: 2012 standards. The laboratory personnel were blinded to the group assignments and patient information. The primary outcome measure was the change in renin (REN) levels during outpatient visits at baseline, 1 month, and 3 months. The secondary outcome was the change in other metabolic biomarkers from baseline to the 3-month visit. To monitor treatment adherence and manage potential side effects, a safety assessment was conducted based on adverse event reports during the study.

## 2.4. Statistical analysis

SPSS 21.0 software was used for the statistical analysis. Continuous variables that followed normal distribution were expressed as mean ± standard deviation, and continuous variables that followed non-normal distribution were presented as median with interquartile range. Based on normally or non-normally distributed variables, the independent samples *t*-test and Mann-Whitney *U* test was employed to compare the two groups, while the paired *t*-test and Wilcoxon matched pairs signed rank test was utilized to compare the corresponding indicators before and after treatment. Statistical significance was set at $p < 0.05$. The multiplicity across multiple timepoints was adjusted using the Bonferroni test, with the significance level adjusted to 0.017. We calculated that sample size of 64 was necessary to provide up to 80% power to detect a difference in mean REN level between two groups, assuming a mean difference of 0.5 ng/L and standard deviation of 1.0 ng/L according to previous study [24] with significance of 0.05 and power of 80%. Hence, we set the sample size to 147.

Spearman correlation analysis was conducted to examine the factors influencing the RAAS. The non-normally distributed variables (AII, ALD, REN, ARR, UACR, CP, HOMA-β, HOMA-IR) were log-transformed, and multiple linear regression was then used to identify the independent factors affecting the RAAS in the two models. Sequential multivariable adjustment modeling was performed:

Model 1: age, total cholesterol, triglycerides, high-density lipoprotein cholesterol, low-density lipoprotein cholesterol, urea nitrogen, creatinine, potassium, sodium, calcium, phosphorus, systolic blood pressure, diastolic blood pressure, body mass index and diabetic duration.

Model 2: Model 1 + the RAAS components (Log-REN, ALD or AII).

## 3. Results

### 3.1. Baseline characteristics between DKD and non-DKD patients

Patients with DKD exhibited significantly elevated levels of FBG (10.68 vs. 9.40 mmol/L), TG (2.13 vs. 1.58 mmol/L), Crea (70.39 vs. 60.26 μmol/L), and UACR (87.55 vs. 10.19 mg/g), alongside reduced HOMA-β (35.00% vs. 43.00%) values

compared with non-DKD patients at the start of the study ($p<0.05$, Table 1). The AII (109 vs. 114 ng/L), ALD (137 vs. 135 ng/L), REN (11.10 vs. 14.00 ng/L), and ARR (1.24 vs. 0.98) values as well as the remaining indicators were not statistically different between the DKD and non-DKD groups at baseline($p>0.05$, Table 1).

### 3.2. The influence of SGLT-2is treatment on biochemical indicators in DKD and non-DKD patients

After treatment with dapagliflozin for 3 months, the BMI (−0.87 vs. −0.93 kg/m²; DKD vs. non-DKD), HbA1c (−1.62% vs. −1.30%), FBG (−2.64 vs. −1.70 mmol/L), TG (−0.88 vs. −0.72 mmol/L), TCHO (−0.41 vs. −0.49 mmol/L), SBP (−9.03 vs. −6.22 mmHg), DBP (−3.47 vs. −3.66 mmHg), UACR (−355.01 vs. −4.66 mg/g), and HOMA-IR (−0.44 vs. −0.23) levels were significantly decreased, while Crea (2.90 vs. 4.14 μmol/L) and HOMA-β (15.70% vs. 11.87%) were significantly increased from the baseline in both DKD and non-DKD groups ($p<0.05$, Table 2). Additionally, the differences from the baseline in FBG and UACR values were significantly higher in DKD group than those in non-DKD group ($p<0.05$, Table 2). S1 and S2 Tables presented the specific biochemical indicator values before and after treatment for both groups. The results of the statistical analysis were consistent with those shown in Table 2.

**Table 1. Baseline characteristics between DKD and non-DKD patients.**

| Variables | DKD | non-DKD | p value |
|---|---|---|---|
| age (y) | 53.27±14.41 | 55.76±12.17 | 0.261 |
| Diabetic duration (M) | 81.0(22.0~213.5) | 72.0(12.0~135.0) | 0.101 |
| BMI (kg/m²) | 26.91(24.94~31.22) | 26.72(24.41~29.26) | 0.297 |
| HbA1c (%) | 9.47±1.82 | 9.08±2.29 | 0.247 |
| FBG (mmol/L) | 10.68±3.12 | 9.40±2.69 | 0.009 |
| TG (mmol/L) | 2.13(1.33~3.87) | 1.58(1.15~2.62) | 0.025 |
| TCHO (mmol/L) | 4.90±1.07 | 4.86±1.34 | 0.848 |
| HDL (mmol/L) | 1.10±0.26 | 1.07±0.29 | 0.458 |
| LDL (mmol/L) | 2.86±0.95 | 2.93±1.09 | 0.677 |
| Urea (mmol/L) | 6.11±2.21 | 5.55±1.31 | 0.063 |
| Crea (μmol/L) | 70.39±28.78 | 60.26±15.26 | 0.008 |
| K (mmol/L) | 4.06±0.27 | 4.06±0.46 | 0.993 |
| Na (mmol/L) | 140.56±2.70 | 140.84±2.62 | 0.534 |
| Ca (mmol/L) | 2.33±0.14 | 2.33±0.13 | 0.816 |
| P (mmol/L) | 1.19±0.19 | 1.19±0.15 | 0.948 |
| SBP (mmHg) | 156.9±21.51 | 153.07±18.02 | 0.243 |
| DBP (mmHg) | 93.19±10.86 | 92.86±11.05 | 0.857 |
| UACR (mg/g) | 87.55(47.34~215.80) | 10.19(6.10~14.81) | <0.001 |
| AII (ng/L) | 109(96~124) | 114(103~123) | 0.204 |
| ALD (ng/L) | 137(98~190) | 135(103~165) | 0.941 |
| REN (ng/L) | 11.10(4.85~22.85) | 14.00(7.33~25.78) | 0.196 |
| ARR | 1.24(0.79~2.25) | 0.98(0.56~1.68) | 0.086 |
| CP (nmol/L) | 0.65(0.38~1.00) | 0.69(0.43~0.91) | 0.768 |
| HOMA-β (%) | 35.00(22.85~47.75) | 43.00(27.85~58.10) | 0.042 |
| HOMA-IR | 2.03(1.08~3.02) | 1.86(1.08~2.49) | 0.329 |

Data are expressed as "mean ± standard deviation" or "median with interquartile range". DKD, diabetic kidney disease; HbA1c, Hemoglobin A1c; FBG, fast blood glucose; TCHO, total cholesterol; HDL, high-density lipoprotein cholesterol; LDL, low-density lipoprotein cholesterol; Urea, urea nitrogen; Crea, creatinine; K, potassium; Na, sodium; Ca, calcium; P, phosphorus; SBP, systolic blood pressure; DBP, diastolic blood pressure; AII, angiotensin II; ALD, aldosterone; REN, renin; ARR, aldosterone-to-renin ratio; CP, C-peptide; HOMA-β, Homeostasis Model Assessment of β-cell function; HOMA-IR, Homeostasis Model Assessment of insulin resistance; UACR, urinary albumin-to-creatinine ratio; TG, triglycerides; BMI, body mass index.

**Table 2. Changes of physical and biochemical indicators from baseline and difference between DKD and non-DKD patients.**

| Δ Variable | DKD | non-DKD | p value |
|---|---|---|---|
| BMI (kg/m²) | −0.87±0.57*ᵃ | −0.93±0.67*ᵃ | 0.543 |
| HbA1c (%) | −1.62±1.41* | −1.30±1.42* | 0.183 |
| FBG (mmol/L) | −2.64±2.66* | −1.70±1.92* | 0.016 |
| TG (mmol/L) | −0.88±1.11*ᵃ | −0.72±0.97*ᵃ | 0.353 |
| TCHO (mmol/L) | −0.41±1.13* | −0.49±1.30* | 0.700 |
| HDL (mmol/L) | 0.01±0.21 | 0.02±0.27 | 0.787 |
| LDL (mmol/L) | −0.06±1.18 | −0.20±0.97 | 0.420 |
| Urea (mmol/L) | 0.25±2.00 | 0.14±1.54 | 0.697 |
| Crea (μmol/L) | 2.90±30.18 | 4.14±16.24* | 0.758 |
| K (mmol/L) | 0.03±0.54 | 0.02±0.52 | 0.964 |
| Na (mmol/L) | −0.16±3.27 | −0.03±3.63 | 0.810 |
| Ca (mmol/L) | 0.00±0.21 | 0.00±0.18 | 0.997 |
| P (mmol/L) | −0.01±0.24 | −0.01±0.17 | 0.953 |
| SBP (mmHg) | −9.03±19.54* | −6.22±16.22* | 0.344 |
| DBP (mmHg) | −3.47±11.05* | −3.66±11.99* | 0.918 |
| UACR (mg/g) | −355.01±1534.12*ᵃ | −4.46±7.86*ᵃ | 0.050 |
| CP (nmol/L) | −0.05±0.20ᵃ | −0.03±0.23ᵃ | 0.513 |
| HOMA-β (%) | 15.70±17.10*ᵃ | 11.87±14.68*ᵃ | 0.148 |
| HOMA-IR | −0.44±0.85*ᵃ | −0.23±0.69*ᵃ | 0.103 |

Data are expressed as "mean ± standard deviation". SGLT-2is, sodium-glucose cotransporter-2 inhibitors; BMI, body mass index; HbA1c, Hemoglobin A1c; FBG, fast blood glucose; TG, triglycerides; TCHO, total cholesterol; HDL, high-density lipoprotein cholesterol; LDL, low-density lipoprotein cholesterol; Urea, urea nitrogen; Crea, creatinine; K, potassium; Na, sodium; Ca, calcium; P, phosphorus; SBP, systolic blood pressure; DBP, diastolic blood pressure; UACR, urinary albumin-to-creatinine ratio; CP, C-peptide; HOMA-β, Homeostasis Model Assessment of β-cell function; HOMA-IR, Homeostasis Model Assessment of insulin resistance. S1 and S2 Tables presented the specific values and the results of the statistical analysis before and after treatment for both groups. Asterisk denotes within-group change vs. baseline (ᵃ, Wilcoxon matched pairs signed rank test; the other rows: paired $t$-test); the $p$-value column compares the difference between DKD and non-DKD (two-sample $t$-test). (*$p < 0.05$ baseline vs. 3 months)

As mentioned above, the DKD group exhibited significantly elevated levels of TG (2.13 vs. 1.58 mmol/L), Crea (70.39 vs. 60.26 μmol/L), and UACR (87.55 vs. 10.19 mg/g) compared with the non-DKD group at the baseline ($p < 0.05$, Table 1). After 3 months of dapagliflozin administration, the levels of TG (1.51 vs. 1.24 mmol/L), Crea (73.29 vs. 64.39 μmol/L), and UACR (38.30 vs. 5.30 mg/g) in DKD group remained significantly higher than those in non-DKD group ($p < 0.05$, S3 Table).

### 3.3. The influence of SGLT-2is treatment on RAAS during the time course

The absolute values of AII, ALD, and REN did not differ statistically between the DKD and non-DKD groups at each time point, including the baseline ($p > 0.05$, Table 3, Fig 3). REN levels increased significantly at the 1-month visit in both the DKD and non-DKD groups ($p < 0.001$ and $p = 0.006$, respectively) (Fig 3C). However, the increase in REN levels from baseline to 1 month was significantly greater in the DKD group compared to the non-DKD group (4.15±7.35 vs. 2.75±8.03 ng/L; DKD vs. non-DKD; $p < 0.05$). After three months, REN levels showed no significant change from the baseline, indicating a return to initial values. However, no significant differences were observed in AII and ALD levels before and after 1- or 3-month treatments in either patient group ($p > 0.05$, Fig 3, Table 3).

Unlike renin, ARR values decreased significantly from baseline at both the 1- month visit ($p < 0.001$) and 3-month visit (DKD, $p = 0.016$; non-DKD, $p = 0.015$) in both patient groups (Table 3 and Fig 3D). Moreover, the differences from baseline

**Table 3. Time Course of the RAAS.**

| Variables | baseline | 1 month | 3 months |
|---|---|---|---|
| DKD | | | |
| AII (ng/L) | 109(96~124) | 102(96~119) | 107(97~121) |
| ALD (ng/L) | 137(98~190) | 135(102~181) | 141(105~173) |
| REN (ng/L) | 11.10(4.85~22.85) | 15.70(7.70~26.85) *** | 13.90(6.85~22.85) ### |
| ARR | 1.24(0.79~2.25) | 0.86(0.65~1.38) *** | 1.15(0.75~1.77) *### |
| non-DKD | | | |
| AII (ng/L) | 114(103~123) | 109(99~125) | 115(103~121) |
| ALD (ng/L) | 135(103~165) | 133(105~161) | 138(110~158) |
| REN (ng/L) | 14.00(7.33~25.78) | 16.90(12.08~26.25) ** | 14.35(9.68~21.83) ### |
| ARR | 0.98(0.56~1.68) | 0.73(0.56~0.94) *** | 0.88(0.63~1.35) *### |

Data are expressed as "median with interquartile range". DKD, diabetic kidney disease; AII, angiotensin II; ALD, aldosterone; REN, renin; ARR, aldosterone-to-renin ratio. (*$p < 0.017$ vs. baseline, **$p < 0.01$ vs. baseline, *** $p < 0.001$ vs. baseline, ### $p < 0.001$ vs. 1 month)

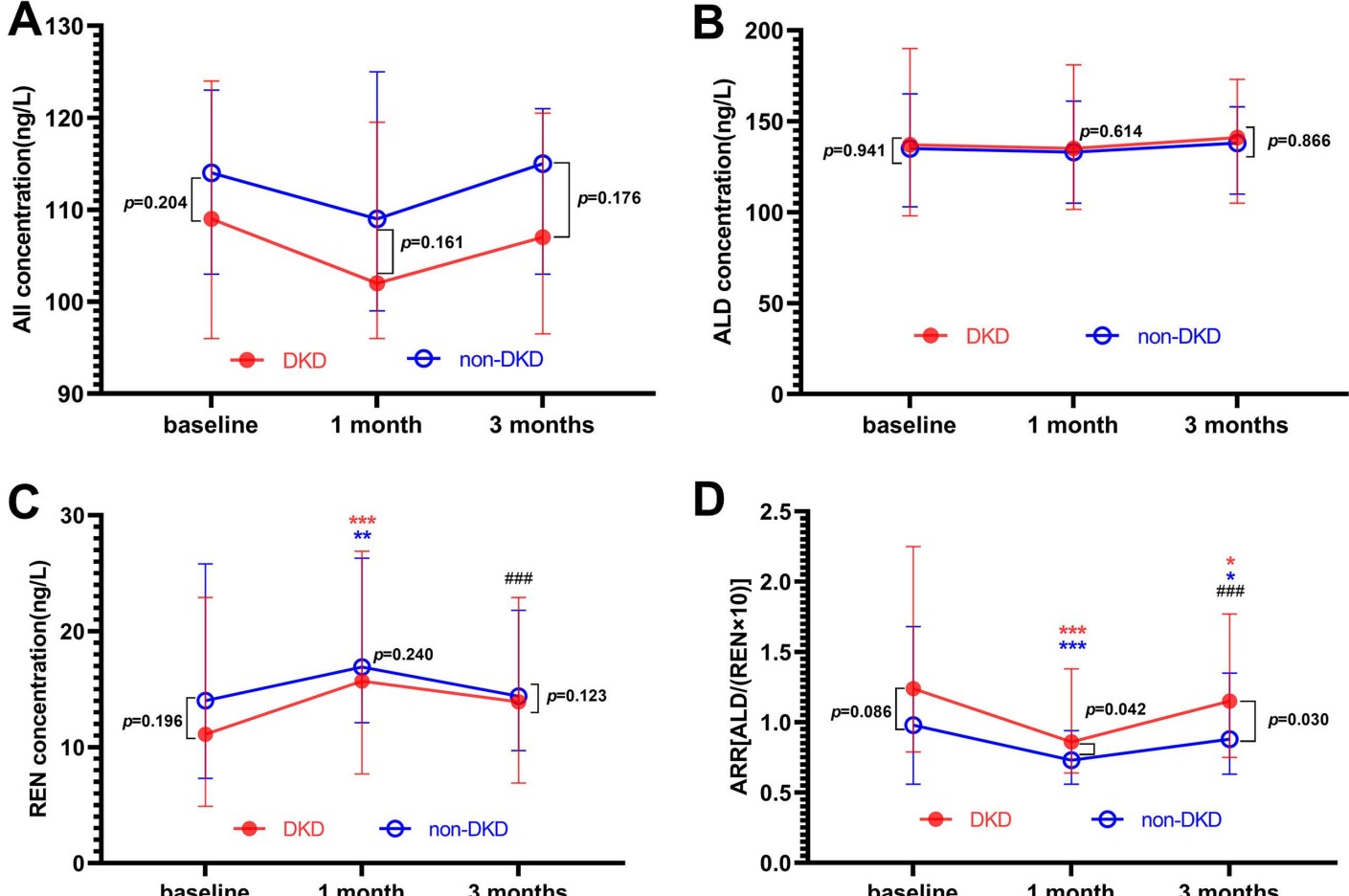

**Fig 3. Effects of SGLT-2is during the time course. (A) Changes in angiotensin II; (B) Changes in aldosterone; (C) Changes in renin; (D) Changes in aldosterone-to-renin ratio.** Nonparametric $p$-values for within-group differences from the baseline are indicated in colored characters and between-group differences in black letters for each timepoint. DKD, diabetic kidney disease. Data are expressed as "median with interquartile range". (*$p < 0.017$ vs. baseline, **$p < 0.01$ vs. baseline, *** $p < 0.001$ vs. baseline, ### $p < 0.001$ vs. 1 month).

to the 1- or 3-month visits were not significant between the two groups (1 month, −0.70±1.16 vs. −0.59±1.19; 3 months, −0.45±1.15 vs. −0.42±1.07; DKD vs. non-DKD; $p > 0.05$). It is noteworthy that there observed no temporal difference for ALD levels, suggesting that ARR was primarily influenced by REN. Interestingly, REN levels returned to baseline after three months of treatment, yet the ARR remained below its initial value.

### 3.4. Spearman correlation analysis of the RAAS and glucose metabolism indicators before and after treatment

In order to investigate the interplay between SGLT-2is, RAAS, and glucose metabolism, a correlation matrix analysis was conducted involving 10 variables: AII, ALD, REN, ARR, HbA1c, FBG, CP, HOMA-β, HOMA-IR and UACR, both before and after 3 months of treatment. The correlation heatmap was presented in Fig 4.

Fig 4 illustrated that, regardless of dapagliflozin treatment, ALD was positively correlated with REN ($r = 0.497$ to $0.757$, $p < 0.05$), while ARR was negatively correlated with REN ($r = −0.955$ to $−0.889$, $p < 0.05$). These findings suggested that ALD and ARR were significantly influenced by the internal interactions of the RAAS.

In the DKD group (Fig 4A), ALD and REN exhibited positive correlations with CP ($r = 0.400$ and $0.414$) and HOMA-IR (($r = 0.467$ and $0.466$), while demonstrating negative correlations with UACR ($r = −0.380$ and $−0.593$). In contrast to REN, ARR displayed negative correlations with CP($r = −0.332$), and HOMA-IR ($r = −0.363$), but positive correlation with UACR ($r = 0.545$). After three months of dapagliflozin administration, these correlations were slightly attenuated but remained statistically significant ($p < 0.05$, Fig 4B).

In the non-DKD group (Fig 4C), ALD and REN exhibited positive correlations with FBG ($r = 0.368$ and $0.545$), CP ($r = 0.373$ and $0.347$), and HOMA-IR ($r = 0.437$ and $0.453$), while REN displayed a negative correlation with UACR ($r = −0.257$). In contrast to REN, ARR demonstrated negative correlations with FBG ($r = −0.498$), CP ($r = −0.270$), and HOMA-IR ($r = −0.370$), but positive correlation with UACR ($r = 0.253$). After three months of dapagliflozin treatment, these correlations were markedly diminished and no longer statistically significant ($p > 0.05$, Fig 4D).

### 3.5. Multiple linear regression analysis of the RAAS and glucose metabolism indicators before and after treatment

Multiple linear regression analysis was conducted using AII, ALD, REN, and ARR as dependent variables, with factors significantly associated with RAAS levels as independent variables in two covariate models. Model 1 included known risk factors for T2DM, such as age, BMI, lipid profiles, creatinine levels, potassium, and hypertension. Model 2 incorporated all variables from Model 1, along with RAAS components, to adjust for their internal interactions. Subsequently, efficacy subgroups were established to assess the effects of dapagliflozin. The findings for each group were presented in Table 4.

In the DKD cohort, after additional adjustment for Log-REN and Log-AII (Model 2), a 100% increase in Log-ALD was associated with a 0.143 nmol/L higher CP (95% CI: 0.002~0.283, $p = 0.047$) and a 15.8% higher HOMA-IR (95% CI: 2.5%~29.0%, $p = 0.020$). After 3 months of treatment, these correlations were markedly diminished and no longer statistically significant ($p = 0.582$ and $0.634$, respectively; Model 2). A 100% increase in Log-REN was associated with a 0.359 mg/g lower UACR (95% CI: 0.210~0.507, $p < 0.001$; Model 2). After treatment, this correlation was slightly attenuated but remained statistically significant (0.124[0.015~0.233] mg/g, $p = 0.027$) (Model 2).

In the non-DKD cohort, after further adjustment for Log-REN (Model 2), the correlations between AII, ALD, ARR, and glucose metabolism indicators were substantially reduced and were no longer statistically significant ($p > 0.05$ for all; Model 2). However, a 100% increase in Log-REN was associated with a 0.042 mmol/L higher FBG (95% CI: 0.012~0.071, $p = 0.007$; Model 2). After treatment, this association was significantly attenuated (0.014 [95% CI: −0.038~0.066] mmol/L, $p = 0.589$) (Model 2).

### 3.6. Adverse events during the study

During the study period, a total of 23 patients (15.6%) experienced 31 adverse events, including 11 cases of thirst, 4 cases of hypoglycemia, 5 cases of genital itching, 6 cases of constipation, 3 cases of skin itching, and 2 cases of pollakiuria. Importantly, no adverse events were classified as severe. A total of 147 patients successfully completed the trial.

| | | AII | ALD | REN | ARR | HbA1c | FBG | CP | β(%) | IR | UACR |
|---|---|---|---|---|---|---|---|---|---|---|---|
| **A** | AII | 1.000 | -0.088 | -0.030 | 0.047 | -0.117 | 0.006 | 0.018 | 0.026 | 0.004 | -0.007 |
| | ALD | -0.088 | 1.000 | 0.679* | -0.355* | 0.045 | 0.227 | 0.400* | 0.132 | 0.467* | -0.380* |
| | REN | -0.030 | 0.679* | 1.000 | -0.909* | -0.045 | 0.238* | 0.414* | 0.168 | 0.466* | -0.593* |
| | ARR | 0.047 | -0.355* | -0.909* | 1.000 | 0.061 | -0.207 | -0.332* | -0.146 | -0.363* | 0.545* |
| **B** | AII | 1.000 | -0.083 | -0.026 | 0.015 | -0.038 | -0.117 | 0.032 | 0.095 | 0.007 | 0.025 |
| | ALD | -0.083 | 1.000 | 0.757* | -0.440* | -0.063 | 0.039 | 0.395* | 0.248 | 0.411* | -0.261* |
| | REN | -0.026 | 0.757* | 1.000 | -0.899* | -0.228 | 0.008 | 0.397* | 0.282 | 0.413* | -0.400* |
| | ARR | 0.015 | -0.440* | -0.899* | 1.000 | 0.267 | -0.016 | -0.287* | -0.205 | -0.300* | 0.422* |
| **C** | AII | 1.000 | 0.299* | 0.266* | -0.192 | 0.283* | 0.302* | 0.116 | -0.128 | 0.167 | -0.011 |
| | ALD | 0.299* | 1.000 | 0.694* | -0.472* | 0.152 | 0.368* | 0.373* | 0.004 | 0.437* | -0.092 |
| | REN | 0.266* | 0.694* | 1.000 | -0.955* | 0.233* | 0.545* | 0.347* | -0.184 | 0.453* | -0.257* |
| | ARR | -0.192 | -0.472* | -0.955* | 1.000 | -0.205 | -0.498* | -0.270* | 0.223 | -0.370* | 0.253* |
| **D** | AII | 1.000 | 0.114 | 0.227 | -0.170 | 0.167 | 0.286 | -0.030 | -0.223 | 0.024 | -0.109 |
| | ALD | 0.114 | 1.000 | 0.497* | -0.099 | 0.101 | 0.168 | 0.169 | -0.031 | 0.182 | 0.234 |
| | REN | 0.227 | 0.497* | 1.000 | -0.889* | 0.099 | 0.345* | 0.202 | -0.124 | 0.243 | -0.058 |
| | ARR | -0.170 | -0.099 | -0.889* | 1.000 | -0.040 | -0.287 | -0.141 | 0.121 | -0.181 | 0.144 |

**Fig 4. Spearman correlation analysis of the RAAS and glucose metabolism indicators before and after treatment in DKD and non-DKD patients. (A)** Pre-treatment in DKD group; **(B)** Post-treatment in DKD group; **(C)** Pre-treatment in non-DKD group; **(D)** Post-treatment in non-DKD group. Heat-map with background in three-color scale where −1 = blue, + 1 = red and 0 = white. DKD, diabetic kidney disease; AII, Angiotensin II; ALD, aldosterone; REN, renin; ARR, aldosterone-to-renin ratio; HbA1c, hemoglobin A1c; FBG, fast blood glucose; CP, C-peptide; β (%), Homeostatic Model Assessment of β-cell function; IR, Homeostatic Model Assessment of Insulin Resistance; UACR, urinary albumin-to-creatinine ratio. (*$p < 0.05$).

## 4. Discussion

The effect of sodium-glucose cotransporter-2 inhibitors (SGLT-2is) on the renin-angiotensin-aldosterone system (RAAS) in type 2 diabetes mellitus (T2DM) remains controversial [12], as they may either activate or inhibit RAAS, potentially impacting the accuracy of ARR in diagnosing primary aldosteronism (PA) [7]. The role of SGLT-2is in reducing blood

**Table 4. Multiple linear regression analysis of the RAAS and glucose metabolism indicators before and after treatment in DKD and non-DKD patients.**

| Pre-treatment in DKD group | | Model 1 | | Model 2 | |
|---|---|---|---|---|---|
| Dependent variables | Independent variables | β | p | β | p |
| Log-ALD | Log-CP | 0.314(0.157~0.471) | <0.001 | 0.143(0.002~0.283) | 0.047 |
| | Log-HOMA-IR | 0.325(0.185~0.466) | <0.001 | 0.158(0.025~0.290) | 0.020 |
| | Log-UACR | −0.134(−0.215~−0.053) | 0.002 | 0.024(−0.061~0.109) | 0.572 |
| Log-REN | FBG | 0.044(0.007~0.080) | 0.019 | 0.013(−0.017~0.042) | 0.385 |
| | Log-CP | 0.713(0.311~1.115) | 0.001 | 0.234(−0.132~0.601) | 0.205 |
| | Log-HOMA-IR | 0.736(0.373~1.098) | <0.001 | 0.255(−0.096~0.606) | 0.151 |
| | Log-UACR | −0.533(−0.701~−0.364) | <0.001 | −0.359(−0.507~−0.210) | <0.001 |
| Log-ARR | Log-CP | −0.400(−0.729~−0.071) | 0.018 | 0.001(−0.002~0.002) | 0.977 |
| | Log-HOMA-IR | −0.411(−0.712~−0.110) | 0.008 | 0.001(−0.002~0.002) | 0.971 |
| | Log-UACR | 0.400(0.265~0.534) | <0.001 | 0.001(−0.001~0.002) | 0.470 |
| Post-treatment in DKD group | | Model 1 | | Model 2 | |
| Log-ALD | Log-CP | 0.319(0.111~0.527) | 0.003 | 0.001(−0.002~0.003) | 0.582 |
| | Log-HOMA-IR | 0.345(0.138~0.552) | 0.001 | 0.001(−0.002~0.003) | 0.634 |
| Log-REN | Log-CP | 0.599(0.060~1.137) | 0.030 | 0.113(−0.307~0.532) | 0.591 |
| | Log-HOMA-IR | 0.658(0.119~1.197) | 0.018 | 0.121(−0.303~0.545) | 0.570 |
| | Log-UACR | −0.200(−0.342~−0.059) | 0.006 | −0.124(−0.233~−0.015) | 0.027 |
| Log-ARR | Log-HOMA-IR | −0.312(−0.704~0.080) | 0.116 | −0.001(−0.004~0.002) | 0.471 |
| | Log-UACR | 0.149(0.040~0.259) | 0.008 | −0.002(−0.001~0.0004) | 0.526 |
| Pre-treatment in non-DKD group | | Model 1 | | Model 2 | |
| Log-AII | HbA1c | 0.008(0.001~0.016) | 0.051 | 0.007(−0.002~0.015) | 0.110 |
| | FBG | 0.007(−0.001~0.014) | 0.072 | 0.005(−0.003~0.014) | 0.203 |
| Log-ALD | FBG | 0.017(0.002~0.031) | 0.024 | −0.003(−0.015~0.009) | 0.588 |
| | Log-CP | 0.145(0.038~0.252) | 0.009 | 0.099(−0.016~0.214) | 0.091 |
| | Log-HOMA-IR | 0.204(0.051~0.358) | 0.010 | 0.095(−0.019~0.210) | 0.101 |
| Log-REN | HbA1c | 0.038(−0.007~0.083) | 0.096 | 0.016(−0.017~0.050) | 0.338 |
| | FBG | 0.070(0.033~0.108) | <0.001 | 0.042(0.012~0.071) | 0.007 |
| | Log-CP | 0.307(−0.150~0.765) | 0.184 | −0.048(−0.394~0.298) | 0.783 |
| | Log-HOMA-IR | 0.429(−0.011~0.869) | 0.056 | 0.025(−0.319~0.369) | 0.883 |
| | Log-UACR | −0.345(−0.699~0.008) | 0.055 | −0.280(−0.532~−0.031) | 0.028 |
| Log-ARR | FBG | −0.054(−0.084~−0.024) | 0.001 | 0.001(−0.002~0.004) | 0.655 |
| | Log-CP | −0.130(−0.494~0.234) | 0.477 | 0.001(−0.001~0.001) | 0.517 |
| | Log-HOMA-IR | −0.224(−0.577~0.128) | 0.208 | 0.001(−0.002~0.005) | 0.425 |
| | Log-UACR | 0.312(0.038~0.587) | 0.027 | 0.001(−0.002~0.003) | 0.784 |
| Post-treatment in non-DKD group | | Model 1 | | Model 2 | |
| Log-REN | FBG | 0.042(−0.020~0.104) | 0.183 | 0.014(−0.038~0.066) | 0.589 |

Model 1: age, total cholesterol, triglycerides, high-density lipoprotein cholesterol, low-density lipoprotein cholesterol, urea nitrogen, creatinine, potassium, sodium, calcium, phosphorus, systolic blood pressure, diastolic blood pressure, body mass index and diabetic duration. Model 2(aldosterone analyses): Model 1 + log-renin and log-angiotensin II. Model 2(renin analyses): Model 1 + log-aldosterone and log-angiotensin II. Model 2(ARR analyses): Model 1 + log-aldosterone, log-renin and log-angiotensin II. Model 2(AII analyses): Model 1 + log-aldosterone and log-renin. ALD, aldosterone; REN, renin; ARR, aldosterone-to-renin ratio; AII, angiotensin II; HbA1c, Hemoglobin A1c; FBG, fast blood glucose; UACR, urinary albumin-to-creatinine ratio; CP, C-peptide; HOMA-IR, Homeostasis Model Assessment of insulin resistance; HOMA-β, Homeostasis Model Assessment of β-cell function.

glucose levels and protecting pancreatic β-cells and renal function aligns with the target organs involved in the RAAS [1]. However, the interplay between SGLT-2is, RAAS, and glucose metabolism has not yet been reported. Therefore, we conducted this prospective study and found that ALD and REN exhibited positive correlations with insulin resistance and hyperglycemia, and negative correlations with UACR in T2DM patients with hypertension. Moreover, the correlations between RAAS markers and glucose metabolism indices were significantly attenuated after SGLT-2is treatment. However, short-term therapy may elevate REN levels and reduce ARR, potentially resulting in false-negative outcomes in PA screening.

SGLT-2is are a novel class of glucose-lowering medications that provide cardio-renal protective effects [9,11]. By inhibiting sodium-coupled glucose reabsorption in the proximal convoluted tubule, they increase the distal delivery of sodium to the macula densa [10,11]. This process not only restores the tubulo-glomerular feedback mechanism and reduces glomerular hyperfiltration and proteinuria [10,11], but also decreases renin levels, functioning similarly to RAAS inhibition [11], thereby slowing the progression of diabetic kidney disease (DKD) [25]. However, the osmotic diuresis and natriuretic effects of SGLT-2is may activate RAAS to maintain normal glomerular filtration rates [11]. Consequently, the association between SGLT-2is and RAAS activation remains controversial. A retrospective observational study indicated that there was no significant effect on ALD, REN levels, or ARR following 2–6 months of SGLT-2is therapy [13]. Conversely, another prospective longitudinal study reported a reduction in ARR after 198 days of SGLT-2is therapy, associated with increased REN levels, while ALD levels remained unchanged [10]. Furthermore, additional evidence suggested that SGLT-2is therapy led to an early increase in REN levels during the initial treatment phase (≤3 months) [12], whereas both REN and ALD levels stabilized with long-term use (>3 months) [12,26], thereby corroborating the transient diuretic effect of SGLT-2is [11,26]. This heterogeneity arose from variations in study designs (e.g., inclusion criteria, follow-up periods), control conditions (such as self-control), diets (including sodium and protein intake), sampling strategies (e.g., consecutive, random, or convenience sampling), medication dosages, and treatment adherence [10,13,25].

Our study suggested that REN levels increased significantly from baseline at 1-month visit, while ARR values decreased significantly from baseline at 1-month and 3-month visits in both groups. Interestingly, no temporal differences were observed in ALD levels. REN levels returned to baseline after three months of treatment; yet the ARR, which was primarily influenced by REN, remained below its initial value. This result indicated that the ARR, calculated using ALD and REN levels, was more sensitive to the effects of drug treatment. A slight increase in renin resulted in a significant decrease in the ARR. Consequently, clinicians should exercise caution when interpreting ARR in T2DM patients undergoing short-term SGLT-2is therapy due to the risk of false-negative PA screening. A prospective study using bioimpedance spectroscopy further corroborate our findings, demonstrating that SGLT-2is can reduce extracellular fluid volume similarly to thiazide diuretics, leading to increased REN levels and decreased ARR [27]. However, the long-term effect of SGLT-2is on fluid volume was comparable to that of hydrochlorothiazide [27]. SGLT-2is diuretic properties possessed unique advantages over thiazides and loop diuretics [11]. These inhibitors mitigated blood volume loss through several mechanisms, including the activation of tubulo-glomerular feedback [15] and the translocation of water from the interstitial space to the intravascular compartment [28], and finally diminished stimulation of the RAAS and the sympathetic nervous system [29]. Multiple studies have investigated the timeline for fluid volume recovery following the discontinuation of SGLT-2is. The EMPA-REG outcome study indicated that the estimated glomerular filtration rate (eGFR), an indicator of fluid volume recovery, returned to baseline at a median of 34 days after stopping SGLT-2is [30]. Therefore, discontinuing SGLT-2is for 4–5 weeks may be advisable before measuring the ARR. However, the optimal timing for medication withdrawal prior to ARR requires further validation.

Patients with DKD exhibited significantly elevated levels of FBG, Crea, UACR, and TG, alongside reduced HOMA-β values, compared to non-DKD patient at the start of our study. However, RAAS components did not differ significantly between the two groups, indicating that blood glucose levels, renal function, and diabetes status did not influence RAAS before treatment, thereby ensuring comparable results afterward. After three months of treatment, both groups

displayed increased HOMA-β levels and decreased BMI, HbA1c, FBG, TG, SBP, DBP, UACR, and HOMA-IR levels compared to baseline measurements. These results suggested that SGLT-2is effectively reduced body weight, blood glucose, lipids, blood pressure, and urinary microalbumin levels, particularly in DKD patients, while also improving insulin resistance and β-cell function. Chronic hyperglycemia before treatment may activate the RAAS [1]. In T2DM patients not using SGLT-2is, renin secretion increased with higher HbA1c levels [18]. Our study found that although HbA1c and FBG levels significantly decreased, REN levels increased, suggesting that extracellular volume depletion caused by SGLT-2is stimulated renin release to counterbalance these effects [10]. SGLT-2is did not influence the secretion of C-peptide and lowered blood glucose levels independently of insulin [2], thereby reducing insulin resistance and improving β-cell function. SGLT-2is decelerated β-cell failure by indirectly reducing excessive glucose influx into these cells [2]. Several studies in animal models of diabetes have demonstrated that treatment with SGLT-2is can enhance β-cell proliferation and/or diminish β-cell apoptosis, thereby preserving β-cell mass [31,32]. Furthermore, our findings indicated that TG levels significantly decreased following treatment, likely due to SGLT-2is reducing systemic glucose toxicity, decreasing TG synthesis in the liver, and enhancing TG metabolism [33]. A meta-analysis has demonstrated that SGLT-2is can elevate TCHO, LDL, and HDL levels while concurrently reducing TG levels [34]. However, other studies have reported no significant correlation between SGLT-2is therapy and dyslipidemia risk [35], indicating the need for further research to clarify their impact on lipid profiles. SGLT-2is provide renal protection by reducing body weight, blood pressure, and UACR, as well as slowing the progression of DKD [9,25]. Consequently, the American Diabetes Association recommends them as the preferred glucose-lowering agents for T2DM patients with kidney disease or UACR over 30 mg/g [36].

Several studies not addressing SGLT-2is have demonstrated that RAAS activation was associated with glucose metabolic abnormalities [18], insulin resistance [21], and impaired renal function [19,20]. Our pre-treatment study found that ALD and REN positively correlated with insulin resistance and hyperglycemia, and negatively correlated with UACR. This suggested that chronic hyperglycemia may activate RAAS, which in turn could further aggravate insulin resistance, impair renal function, and exacerbate glucose metabolic abnormalities [1]. Our results are consistent with those of Joseph JJ's study [21], indicating that ALD\REN was associated with glucose metabolism prior to treatment, independently of known risk factors for T2DM, including BMI, creatinine levels, lipid profiles, and hypertension, as well as independently of interactions within RAAS. After three months of SGLT-2is treatment, RAAS markers (AII, REN, and ALD) returned to baseline levels, while glucose-related indicators decreased significantly. These findings suggested that the RAAS-glucose metabolism interplay was significantly attenuated, potentially due to lowered blood glucose levels caused by SGLT-2is treatment [1]. Our analysis revealed that ARR values decreased with elevated FBG, CP, and HOMA-IR levels, and increased with higher UACR levels, but the correlations were not independent of the RAAS system and SGLT-2is treatment. Metabolic indicators (including blood glucose, insulin resistance, and urinary microalbumin levels) that influence both ALD and REN also indirectly impact ARR. Following treatment, reductions in these indicators corresponded with their diminished effects on ARR. The decrease in ARR primarily resulted from the stimulation of renin secretion and release by the kidneys, a response triggered by the reduction in extracellular fluid volume caused by short-term SGLT-2is treatment [10,11]. Therefore, further investigation into the interplay between RAAS and glucose metabolism, both before and after SGLT-2is treatment, would enhance the clinical interpretation of how these medications interfere with ARR.

This study has several potential limitations. Our observations of RAAS component changes were restricted to 3-month dapagliflozin treatment, leaving us without long-term data to identify the optimal duration for discontinuing SGLT-2is, which was essential for minimizing the risk of false-negative results. Secondly, our study did not include patients with PA; instead, we selected participants at risk for PA, specifically those with T2DM and hypertension. This approach provided the opportunity to assess whether SGLT-2is increase the risk of false-negative screening results [10], as most existing researches on drug interference with ARR have been conducted on patients without PA [10,37,38]. Thirdly, we did not

alter other glucose-lowering medications prior to the study, as the available evidence indicated that they had little effect on RAAS. Fourthly, the RAAS indicators were evaluated at only three time points: at baseline, 1month and 3 months. Furthermore, the dynamic changes in ALD and REN in relation to glucose metabolism were not assessed. Finally, this study recruited patients through convenience consecutive sampling at a single center. Despite employing robust statistical techniques, the possibility of residual, unmeasured confounding factors cannot be excluded. Consequently, further prospective multicenter studies with larger sample size are warranted and should include patients with confirmed PA.

## 5. Conclusions

The correlations between RAAS markers and glucose metabolism indices were significantly attenuated, potentially due to lowered blood glucose levels after SGLT-2is treatment. However, short-term therapy (≤3 months) may elevate REN levels and reduce ARR. Consequently, clinicians should exercise caution when interpreting ARR in T2DM patients on short-term SGLT-2is therapy due to the risk of false-negative PA screening. Further studies to determine the optimal duration for discontinuing SGLT-2is before ARR test are warranted and should validate these findings in PA patients.

## Supporting information

**S1 Table. The influence of SGLT-2is treatment on physical and biochemical indicators in DKD patients.**
(DOCX)

**S2 Table. The influence of SGLT-2is treatment on physical and biochemical indicators in non-DKD patients.**
(DOCX)

**S3 Table. Comparison analysis of RAAS and other biochemical indicators between DKD and non-DKD patients after 3-month treatment.**
(DOCX)

## Acknowledgments

We would like to thank the subjects involved in this study and the staff of Department of Laboratory Medicine for their technical support.

## Author contributions

**Conceptualization:** Ningning Wang, Junfeng Kong, Baohong Yue.

**Data curation:** Ningning Wang, Ziming Lu, Erjun Tian.

**Formal analysis:** Ningning Wang, Erjun Tian.

**Funding acquisition:** Fei Cao.

**Methodology:** Ningning Wang, Junfeng Kong, Ziming Lu, Junhui Li, Shutong Li, Baohong Yue.

**Project administration:** Fei Cao.

**Software:** Ningning Wang, Junhui Li, Shutong Li, Shuai Liu.

**Supervision:** Junfeng Kong, Shuai Liu, Baohong Yue.

**Visualization:** Ningning Wang, Baohong Yue.

**Writing – original draft:** Ningning Wang.

**Writing – review & editing:** Ningning Wang, Junfeng Kong, Ziming Lu, Fei Cao, Erjun Tian, Junhui Li, Shutong Li, Shuai Liu, Baohong Yue.

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
