## [Decision Letter · Decision Letter 0]

2 Jun 2025

Dear Dr. Wang,

Thank you for submitting your manuscript to PLOS ONE. After careful consideration, we feel that it has merit but does not fully meet PLOS ONE’s publication criteria as it currently stands. Therefore, we invite you to submit a revised version of the manuscript that addresses the points raised during the review process.

We look forward to receiving your revised manuscript.

Kind regards,

Daniele Romanello

Academic Editor

PLOS ONE

 [This study was supported by the National Natural Science Foundation of China (grant numbers 82474325). Author FC is a participant in the grant program and contributed to project administration and manuscript review. The other authors have no relevant competing interests.]. 

Additional Editor Comments:

Dear Author,

Sorry for the delay in the response. Unfortunately there was a big discrepancy between the final evaluation of the reviewer and I decided to wait for a 3rd.

I decided to give you the opportunity to upgrade your paper following the reviewers indication and I'm sure that the revision will be exhaustive.

Reviewers' comments:

Reviewer's Responses to Questions

**Comments to the Author**

1. Is the manuscript technically sound, and do the data support the conclusions?

Reviewer #1: Yes

Reviewer #2: Yes

Reviewer #3: Partly

2. Has the statistical analysis been performed appropriately and rigorously?

Reviewer #1: I Don't Know

Reviewer #2: Yes

Reviewer #3: Yes

3. Have the authors made all data underlying the findings in their manuscript fully available?

Reviewer #1: Yes

Reviewer #2: Yes

Reviewer #3: Yes

4. Is the manuscript presented in an intelligible fashion and written in standard English?

Reviewer #1: Yes

Reviewer #2: Yes

Reviewer #3: Yes

Reviewer #1: SGLT2i are cornerstone of the management of T2DM. They are very important tools in delaying the progression of chronic kidney disease and reducing the risk of hospitalization for heart failure. therefore, it is important to understand their mechanism of action and the effect on other pathophysiological mechanisms of complications. the study is an important study.

the number of the studied patients is too smal for such study to make the results credibale. More important it is known that SGLT2i results in early short term rise in serum creatinine and drop of creatinine clearance followed by slowing of the deterioration of kidney function; a study for one month may give results different from long term studies as shown in previous studies.

It is better to use DKD instead of DN which may be interpreted as diabetic neuropathy.

The reference for "diagnostic criteria for diabetic complications" should be mentioned in the methods.

promoting the excretion of glycosuria is better to be replaced by increasing glycosuria.

preventing β-cell failure is a wishful thinking rather than evidence based effect. So it is better to use do not increase brta cell apoptosis (which is an inevitable part of the diabetic syndrome).

how did you diagnose albuminurea, I thing single random test is not enough and 2 positive tests out of three are needed

Reviewer #2: 1. The study addresses an important and underexplored interaction between SGLT-2 inhibitors, RAAS, and glucose metabolism in T2DM with implications for primary aldosteronism (PA) is commendable.

2. Detailed evaluation of multiple metabolic and RAAS components (e.g., ALD, REN, ARR, HOMA indices) provides a robust biochemical profile.

3. A one-month treatment duration is insufficient to draw long-term conclusions, especially regarding ARR variability and its implications for PA screening.

4. Further explanation is needed on how short-term SGLT-2i-induced ARR changes might clinically mislead PA diagnosis.

5. Heatmaps and regression tables should include clearer legends and color scales for easier interpretation

6. This manuscript would benefit from proof reading and simplifying complex sentences improve the manuscript’s readability.

7. While DN and non-DN groups are compared, additional stratification based on baseline ARR or insulin resistance could provide deeper insights

8. There is no mention of statistical power or sample size estimation raises concerns about the adequacy of the sample to detect differences.

9. The availability of raw data in a public repository is asserted, but the actual link or repository is not provided.

Reviewer #3: The authors have done an excellent job of selecting a highly relevant and impactful research topic and has presented it in a clear and engaging manner. This demonstrates both a good understanding of the field and a commitment to addressing an important area of inquiry. However I would like to raise certain queries and thoughts as regards the manuscript

Title: The current title is informative and captures the study’s scope, but its length and vagueness in certain terms (e.g., “exploring the interaction”) reduce its impact. The title does not indicate whether the study is observational, experimental, or prospective, which could help readers quickly understand the study’s methodology and rigor. Refining the title to be more concise, precise, and aligned with the study’s findings—particularly by clarifying the speculative nature of PA implications—will enhance its appeal to readers and better reflect the study’s contribution.

Introduction: The introduction is too lengthy and doesn’t bring out effectively the gap and rationale for this research. It would be suggested that authors could work and incorporate these aspects into introduction to make it more crisp.

Aim and objective : The aim is vague, the objective is overambitious with the claim about PA diagnosis strategies, and there is lack of specificity regarding the SGLT-2i agent. Recommended to revise the aim and objective to be more clearly worded , concise and specific.

Methodology:

Study design - The selection of the study design is congruent with the aim and objective of this research. But the methodology does not clarify the sampling method (e.g., consecutive, random, or convenience sampling), which could introduce selection bias. It is unclear how patients were identified or approached for recruitment. Authors to specify the sampling strategy to ensure reproducibility and address potential biases.

• The criteria for diagnosing diabetic nephropathy are referenced as "established diagnostic criteria" but not explicitly described or cited, which reduces transparency. Additionally, the near-identical median diabetic duration (72 months) in both groups seems unusual and may warrant explanation or verification.

• The rationale for the one-month treatment duration is not justified, and it is unclear whether this duration is sufficient to observe meaningful changes in RAAS or other indicators. Additionally, there is no mention of monitoring adherence to the treatment or managing potential side effects.

• The washout period for medications affecting ARR (2–4 weeks) is broad and not standardized, which could introduce variability. The use of terazosin for poorly controlled blood pressure is mentioned, but the criteria for "poorly controlled" are not defined, and its potential impact on study outcomes is not addressed. Additionally, the exclusion of patients with contraindications to dapagliflozin is vague, as specific contraindications are not listed.

• The timing of post-treatment assessments is not clearly stated (e.g., immediately after the one-month treatment or at a specific follow-up point). Additionally, there is no mention of quality control measures for laboratory assays (e.g., calibration, inter-assay variability) or blinding of laboratory personnel to group allocation.

• The methodology does not specify how sample size was determined, which is critical for assessing the study’s power to detect differences.

• The term “two covariate models” in the regression analysis is unclear, as the covariates included are not specified. Additionally, there is no mention of adjusting for multiple comparisons, which could increase the risk of type I errors given the number of indicators tested.

• There is no mention of a data monitoring committee or procedures for handling adverse events during the study, which are important for patient safety in clinical trials.

Results:

• The text does not provide effect sizes or actual values for the changes in REN and ARR, making it difficult to assess the clinical significance of these findings. Additionally, the lack of comparison between DN and non-DN groups post-treatment (e.g., whether the magnitude of REN increase differed) limits the interpretation of group-specific effects.

• The reporting of statistical significance (e.g., p<0.05) is consistent, but the text does not discuss the clinical relevance of the observed changes (e.g., whether a 1.59% reduction in HbA1c in DN patients is meaningful). The glycosylated hemoglobin was evaluated 1 month after the treatment initiation. While some minor changes in HbA1c may be seen within 4 weeks(lifespan of RBC is 120 days) , it typically takes 8-12 weeks to show some meaningful changes. Hence it is difficult to understand the rationale of testing this parameter within 1 month.

• The mention of significant reductions in Crea (DN group) and TCHO/LDL (non-DN group) is brief and lacks context about their implications. Additionally, the persistent elevation of UACR and TG in the DN group post-treatment is noted but not explored further.

• The tables include variables like age and diabetic duration, which are not expected to change post-treatment, yet diabetic duration shows a statistically significant increase (p<0.001) in both groups, which seems implausible within a one-month study period and suggests a potential error in data reporting or analysis. But if it’s not an error and authors have tried to incorporate the 1 month of study period , I think it will not be sound practice to do so.

• Additionally, the tables are dense, and some variables (e.g., K, Na, Ca, P) show no significant changes, which could be streamlined to focus on key findings.

• The text lists multiple correlations but does not provide correlation coefficients (e.g., r values) or their magnitudes, making it difficult to assess the strength of relationships. The loss of certain correlations post-treatment (e.g., ALD and UACR in DN) is noted but not discussed in terms of biological or clinical implications. It would be advisable to report specific correlation coefficients (r values) for significant findings to quantify the strength of associations. Discuss the biological or clinical relevance of key correlations (e.g., why ALD and REN correlate with CP and HOMA-IR) and the loss of correlations post-treatment.

• The regression results are complex and difficult to interpret due to vague phrasing (e.g., “correlations were not independent of the RAAS and dapagliflozin treatment”). The distinction between Model 1 and Model 2 is not clearly explained in the text, and the rationale for including specific covariates is not provided. Additionally, some p-values in Table 3 are reported as “<0.001” for non-significant results (e.g., Log-ARR in DN pre-treatment), which is confusing and likely a typographical error. It would be advisable to clarify the purpose and differences between Model 1 and Model 2 in the text, specifying which covariates were included and why. Use precise language to describe whether correlations were independent of RAAS components, dapagliflozin, or both.

• Specify whether adjustments for multiple comparisons (e.g., Bonferroni correction) were applied to control for type I errors. Include effect sizes for significant findings to provide context for their magnitude. Briefly explain the implications of log-transformation for regression analyses (e.g., how it affects the interpretation of β coefficients).

Discussion and conclusion:

• The discussion mentions that SGLT-2is "weakened the correlations between RAAS and glucose metabolism disorders," but the specific mechanisms underlying this observation are not clearly elucidated. The authors should provide a more detailed explanation of how these correlations are mitigated and whether this is a direct or indirect effect of SGLT-2is.

• The discussion presents conflicting evidence about SGLT-2is’ effects on RAAS (e.g., no significant effect on aldosterone [ALD] or REN in some studies [17], versus increased REN and reduced ARR in others [14, 16]). While this reflects the controversy, the authors do not sufficiently reconcile these discrepancies or propose hypotheses for why these differences exist (e.g., differences in study design, patient populations, or SGLT-2i agents).

• The study focuses on short-term (1-month) effects of dapagliflozin, but the discussion only briefly mentions that REN and ALD levels stabilize with long-term use (>3 months). A more robust discussion of how short-term versus long-term effects might influence clinical decision-making (e.g., timing of PA screening) would strengthen the manuscript.

• While limitations are acknowledged, the discussion does not adequately address their implications. For example, the lack of PA patients is a significant limitation, as the study’s findings on ARR and false-negative PA screening are speculative without direct evidence from PA patients. The authors should discuss how this limits the generalizability of their findings.

• The claim that SGLT-2is and RAAS antagonists produce synergistic effects [28, 39-41] is not sufficiently supported by the study’s data, as the study did not directly investigate combination therapy. This statement should be tempered or supported with more specific evidence from the current study.

• The conclusion is concise but could be more impactful by summarizing the clinical implications more explicitly. For example, state that “clinicians should be cautious when interpreting ARR in T2DM patients on short-term SGLT-2i therapy due to the risk of false-negative PA screening.”

• Highlight the need for future research to determine the optimal duration for discontinuing SGLT-2is before PA screening and to validate these findings in PA patients.

**Do you want your identity to be public for this peer review?** For information about this choice, including consent withdrawal, please see our Privacy Policy

Reviewer #1: No

Reviewer #2: **Yes: ** Bala Nimmana

Reviewer #3: **Yes: ** Dr Vishakha Jain

---

## [Author Response · Author response to Decision Letter 1]

31 Jul 2025

2025.07.31

Dear Daniele Romanello:

Re: PONE-D-25-15425

Title: Effects of SGLT-2 inhibitors on renin-angiotensin-aldosterone system and their correlation with glucose metabolism in type 2 diabetes mellitus patients with hypertension: A prospective study

Thank you for your kind letter of “PLOS ONE PONE-D-25-15425 Revision Request” on June 02, 2025. We thank the editorial requests for the corrections, and we deeply apologized for the considerable inconvenience caused to you due to our errors exist. We have now revised the manuscript according to the concerns raised in the reviews. A detailed copy of the authors' “point-by-point response to the reviewer's comments” is attached, and specific changes addressing reviewers' comments are marked.

Again, we would like to thank you for taking the time to review our manuscript. We would be most grateful for your kind consideration of this revised manuscript for potential publication in PLOS ONE.

We look forward to hearing from you at your earliest convenience.

Sincerely yours,

Ningning Wang

Corresponding author:

Baohong Yue, Professor, PhD

Address: Department of Laboratory Medicine, The First Affiliated Hospital of Zhengzhou University, Zhengzhou, P. R. China

Responses to the Reviewer’s comments:

Re: PONE-D-25-15425

Title: Effects of SGLT-2 inhibitors on renin-angiotensin-aldosterone system and their correlation with glucose metabolism in type 2 diabetes mellitus patients with hypertension: A prospective study

We would like to acknowledge the reviewer’s insightful comments and valuable suggestions. In addition, we thank the reviewer for helping us improve the present manuscript. We have addressed the concerns raised by the reviewer as detailed below.

Reviewer #1

C1: SGLT2i are cornerstone of the management of T2DM. They are very important tools in delaying the progression of chronic kidney disease and reducing the risk of hospitalization for heart failure. therefore, it is important to understand their mechanism of action and the effect on other pathophysiological mechanisms of complications. the study is an important study.

A1: We are appreciated to note the favorable comments of reviewer in the comments.

C2: the number of the studied patients is too small for such study to make the results credible. More important it is known that SGLT2i results in early short-term rise in serum creatinine and drop of creatinine clearance followed by slowing of the deterioration of kidney function; a study for one month may give results different from long term studies as shown in previous studies.

A2: Great thanks for your comment. We have extended the follow-up period by two months, and “patients were prescribed 10 mg dapagliflozin daily for 3 months. The RAAS and other relevant biomarkers were measured and compared during outpatient visits at baseline, 1 month, and 3 months”. All data, tables, and graphs have been thoroughly reorganized and reanalyzed. The results, abstract, introduction and discussion sections have been revised accordingly.

C3: It is better to use DKD instead of DN which may be interpreted as diabetic neuropathy. The reference for "diagnostic criteria for diabetic complications" should be mentioned in the methods.

A3: Great thanks for your comment. We have used DKD instead of DN, and the entire document has been thoroughly reviewed and revised. The reference [23] for "diagnostic criteria for diabetic complications" has been added to Section 2.1 of the Methods.

C4: promoting the excretion of glycosuria is better to be replaced by increasing glycosuria. preventing β-cell failure is a wishful thinking rather than evidence-based effect. So, it is better to use do not increase beta cell apoptosis (which is an inevitable part of the diabetic syndrome).

A4: Great thanks for your comment. We have revised “promoting the excretion of glycosuria” to “increasing glycosuria” and revised “preventing β-cell failure” to “decelerating β-cell failure”.

C5: how did you diagnose albuminuria, I thing single random test is not enough and 2 positive tests out of three are needed.

A5: Great thanks for your comment. Yes, urinary albumin was evaluated by collecting three midstream urine samples over three consecutive days. The UACR exceeding 30 mg/g in two out of three tests indicated the presence of microalbuminuria.

Reviewer #2

C1: The study addresses an important and underexplored interaction between SGLT-2 inhibitors, RAAS, and glucose metabolism in T2DM with implications for primary aldosteronism (PA) is commendable.

A1: We are appreciated to note the favorable comments of reviewer in the comments.

C2: Detailed evaluation of multiple metabolic and RAAS components (e.g., ALD, REN, ARR, HOMA indices) provides a robust biochemical profile.

A2: We are appreciated to note the favorable comments of reviewer in the comments.

C3: A one-month treatment duration is insufficient to draw long-term conclusions, especially regarding ARR variability and its implications for PA screening.

A3: Great thanks for your comment. We have extended the follow-up period by two months, and “patients were prescribed 10 mg dapagliflozin daily for 3 months. The RAAS and other relevant biomarkers were measured and compared during outpatient visits at baseline, 1 month, and 3 months”. All data, tables, and graphs have been thoroughly reorganized and reanalyzed. The results, abstract, introduction and discussion sections have been revised accordingly.

C4: Further explanation is needed on how short-term SGLT-2i-induced ARR changes might clinically mislead PA diagnosis.

A4: Great thanks for your comment. We have elaborated further explanation in the second and third paragraphs of the Discussion, and the conclusions have been updated accordingly. For example, “However, short-term therapy (≤3 months) may elevate REN levels and reduce ARR. Consequently, clinicians should exercise caution when interpreting ARR in T2DM patients on short-term SGLT-2is therapy due to the risk of false-negative PA screening. Further studies to determine the optimal duration for discontinuing SGLT-2is before PA screening are warranted and should validate these findings in PA patients”.

C5: Heatmaps and regression tables should include clearer legends and color scales for easier interpretation.

A5: Great thanks for your comment. Heatmaps and regression tables have been revised to facilitate easier interpretation.

C6: This manuscript would benefit from proof reading and simplifying complex sentences improve the manuscript’s readability.

A6: Great thanks for your comment. We have carefully revised the text, and an English language editor has edited this study. The revised parts were highlighting in the manuscript.

C7: While DN and non-DN groups are compared, additional stratification based on baseline ARR or insulin resistance could provide deeper insights.

A7: Great thanks for your comment. This article conducted both inter-group horizontal comparisons and longitudinal analyses over time. Additionally, a correlation analysis between RAAS and glucose metabolism indicators was performed. Due to length constraints, further stratification based on baseline ARR and IR was not included. This is a valuable suggestion; however, in this study, an additional comparison of temporal differences was incorporated when comparing DKD and non-DKD groups. The results are as follows: In 3.2 section�“Additionally, the differences from the baseline in FBG and UACR values were significantly higher in DKD group than those in non-DKD group (Table 2).” “After 3 months of dapagliflozin administration, the levels of TG (1.51 vs. 1.24 mmol/L), Crea (73.29 vs. 64.39 µmol/L), and UACR (38.30 vs. 5.30) in DKD group remained significantly higher than those in non-DKD group (p<0.05, S3 Table).” In 3.3 section�“However, the increase in REN levels from baseline to 1 month was significantly greater in the DKD group compared to the non-DKD group (4.15±7.35 vs. 2.75±8.03 ng/L; DKD vs. non-DKD; p<0.05).”

C8: There are no mention of statistical power or sample size estimation raises concerns about the adequacy of the sample to detect differences.

A8: Great thanks for your comment. In the statistical analysis, methods for estimating sample size have been included. The statement is as follows: “We calculated that sample size of 64 was necessary to provide up to 80% power to detect a difference in mean REN level between two groups, assuming a mean difference of 0.5 ng/L and standard deviation of 1.0 ng/L according to previous study with significance of 0.05 and power of 80%. Hence, we set the sample size to 147.”

C9: The availability of raw data in a public repository is asserted, but the actual link or repository is not provided.

A9: Great thanks for your comment. The raw data has been uploaded to the public repository available at https://github.com/ningningwang688/SGLT-2is-on-T2DM.

Reviewer #3

C1: The authors have done an excellent job of selecting a highly relevant and impactful research topic and has presented it in a clear and engaging manner. This demonstrates both a good understanding of the field and a commitment to addressing an important area of inquiry. However, I would like to raise certain queries and thoughts as regards the manuscript.

A1: We are appreciated to note the favorable comments of reviewer in the comments.

C2: Title: The current title is informative and captures the study’s scope, but its length and vagueness in certain terms (e.g., “exploring the interaction”) reduce its impact. The title does not indicate whether the study is observational, experimental, or prospective, which could help readers quickly understand the study’s methodology and rigor. Refining the title to be more concise, precise, and aligned with the study’s findings—particularly by clarifying the speculative nature of PA implications—will enhance its appeal to readers and better reflect the study’s contribution.

A2: Great thanks for your comment. We have revised the title as “Effects of SGLT-2 inhibitors on renin-angiotensin-aldosterone system and their correlation with glucose metabolism in type 2 diabetes mellitus patients with hypertension: A prospective study”

C3: Introduction: The introduction is too lengthy and doesn’t bring out effectively the gap and rationale for this research. It would be suggested that authors could work and incorporate these aspects into introduction to make it more crisp.

A3: Great thanks for your comment. We have revised the introduction, simplified the sentences, and summarized the main points.

C4: Aim and objective: The aim is vague, the objective is overambitious with the claim about PA diagnosis strategies, and there is lack of specificity regarding the SGLT-2i agent. Recommended to revise the aim and objective to be more clearly worded, concise and specific.

A4: Great thanks for your comment. We have revised the objective as “The objective was to accurately evaluate the effects of SGLT-2is on RAAS and glucose metabolism and to improve the clinical interpretation of ARR test.”

Methodology:

C5: Study design - The selection of the study design is congruent with the aim and objective of this research. But the methodology does not clarify the sampling method (e.g., consecutive, random, or convenience sampling), which could introduce selection bias. It is unclear how patients were identified or approached for recruitment. Authors to specify the sampling strategy to ensure reproducibility and address potential biases.

A5: Great thanks for your comment. In 2.1 section of the Method, the sampling strategy have been included. The statement is as follows: “Patients diagnosed with T2DM and hypertension were identified and recruited through convenience consecutive sampling at routine diabetes outpatient visits to the First People's Hospital of Pingdingshan between April 2024 and November 2024.”

C6: The criteria for diagnosing diabetic nephropathy are referenced as "established diagnostic criteria" but not explicitly described or cited, which reduces transparency. Additionally, the near-identical median diabetic duration (72 months) in both groups seems unusual and may warrant explanation or verification.

A6: Great thanks for your comment. The reference [23] for "diagnostic criteria for diabetic complications" has been added to Section 2.1 of the Methods. The median duration of diabetes in DKD group was 81 months. We apologize for our errors exist, and have corrected the inaccurate time.

C7: The rationale for the one-month treatment duration is not justified, and it is unclear whether this duration is sufficient to observe meaningful changes in RAAS or other indicators. Additionally, there is no mention of monitoring adherence to the treatment or managing potential side effects.

A7: Great thanks for your comment. We have extended the follow-up period by two months, and “patients were prescribed 10 mg dapagliflozin daily for 3 months. The RAAS and other relevant biomarkers were measured and compared during outpatient visits at baseline, 1 month, and 3 months”. All data, tables, and graphs have been thoroughly reorganized and reanalyzed. The results, abstract, introduction and discussion sections have been revised accordingly.

We have added the safety assessment. The statements are as follows: In 2.3 section of the Methods: “To monitor treatment adherence and manage potential side effects, a safety assessment was conducted based on adverse event reports during the study.”

In 2.1 section of the Results: “During the study period, a total of 23 patients (15.6%) experienced 31 adverse events, including 11 cases of thirst, 4 cases of hypoglycemia, 5 cases of genital itching, 6 cases of constipation, 3 cases of skin itching, and 2 cases of pollakiuria. Importantly, no adverse events were classified as severe. A total of 147 patients successfully completed the trial.”

C8: The washout period for medications affecting ARR (2–4 weeks) is broad and not standardized, which could introduce variability. The use of terazosin for poorly controlled blood pressure is mentioned, but the criteria for "poorly controlled" are not defined, and its potential impact on study outcomes is not addressed. Additionally, the exclusion of patients with contraindications to dapagliflozin is vague, as specific contraindications are not listed.

A8: Great thanks for your comment. We have revised the vague statements. In 2.2 section: “Patients had not taken any agents that could markedly affect the ARR, such as ACEIs, ARBs, diuretics, or calcium channel blockers within the last 2 months, or aldosterone receptor antagonists, β-blockers, or direct renin inhibitors within the last 4 months [8]; For patients with poorly controlled blood pressure (≥160/100 mmHg), medications with less effect on RAAS, such as α-blockers (e.g., trazodone), were administered.”

We have included the specific contraindications. In 2.2 section: “Patients with contraindications to dapagliflozin therapy (experiencing side effects such as drug hypersensitivity, complicated urinary tract infections or genital infections, and acute kidney injury);”

C9: The timing of post-treatment assessments is not clearly stated (e.g., immediately after the one-month treatment or at a specific follow-up point). Additionally, there is no mention of quality control measures for laboratory assays (e.g., calibration, inter-assay variability) or blinding of laboratory personnel to group allocation.

A9: Great thanks for your comment. We have added the statements. In 2.1 section: “After obtaining written informed consent, eligible patients were prescribed dapagliflozin at a daily dosage of 10 mg for a period of three months. The RAAS and other relevant indicators were measured during outpatient visits at baseline, 1 month, and 3 months.”

In 2.3 section: “All tests were processed and analyzed concurrently with the daily specimens according to ISO (International Organisation for Standardisation) 15189: 2012 standards. The laboratory personnel were blinded to the group assignments and patient information.”

C10: The methodology does not specify how sample size was determined, which is critical for assessing the study’s power to detect differences.

A10: Great thanks for your comment. In the

---

## [Decision Letter · Decision Letter 1]

18 Sep 2025

Dear Dr. Wang,

We look forward to receiving your revised manuscript.

Kind regards,

Zhanjun Jia

Academic Editor

PLOS ONE

Journal Requirements:

Reviewer's Responses to Questions

**Comments to the Author**

Reviewer #2: All comments have been addressed

Reviewer #4: All comments have been addressed

2. Is the manuscript technically sound, and do the data support the conclusions?

Reviewer #2: Yes

Reviewer #4: Partly

3. Has the statistical analysis been performed appropriately and rigorously?

Reviewer #2: Yes

Reviewer #4: No

4. Have the authors made all data underlying the findings in their manuscript fully available?

Reviewer #2: Yes

Reviewer #4: No

5. Is the manuscript presented in an intelligible fashion and written in standard English?

Reviewer #2: Yes

Reviewer #4: No

Reviewer #2: 1. The extension of follow-up from one month to three months substantially strengthens the study’s validity and addresses earlier concerns regarding short-term variability in RAAS parameters.

2. Clarifying the sampling method and recruitment period enhances reproducibility and transparency.

3. The explicit description of inclusion/exclusion criteria and medication washout timelines reduces ambiguity in patient selection.

4. Inclusion of sample size estimation addresses concerns about statistical power.

5. Discuss more explicitly the clinical implications of persistent UACR elevation in DKD patients post-treatment.

6. Consider adding a limitations paragraph addressing residual confounding due to convenience sampling and single-center design.

Reviewer #4: Peer-Review Report – PLOS ONE

Manuscript: PONE-D-25-15425R1

A) Executive Summary and Overall Recommendation

Study focus: Prospective, pre–post evaluation of SGLT-2 inhibitor (dapagliflozin) effects on RAAS components (aldosterone, renin, aldosterone–renin ratio [ARR]) and their relationships with glucose metabolism among adults with type 2 diabetes and hypertension, including DKD vs. non-DKD subgroups.

Main findings: Within 1–3 months of treatment, renin increased, ARR decreased, and glycemic indices improved. Authors warn about potential false-negative primary aldosteronism (PA) screening shortly after initiating SGLT-2 inhibitors.

Verdict: Promising and clinically relevant dataset, but requires major revisions to resolve methodological inconsistencies (treatment duration; medication washout), clarify ARR computation/units, specify statistical models and multiplicity control, ensure consistent terminology, and clean language/formatting.

B) Major Scientific and Statistical Issues (Actionable)

Internal inconsistency in intervention period

Abstract and Results imply a 3‑month protocol, while Methods elsewhere state “one month” and then analyze baseline vs 1 and 3 months. Reconcile everywhere (Abstract, Methods–Intervention, Figure captions, Limitations/Conclusions). State the exact dosing schedule and assessment timepoints (baseline, 1 month, 3 months).

ARR units and computation

ARR is unitless. Remove units (e.g., “ARR −0.59±1.19 ng/L” is incorrect). Specify assays and units for aldosterone and renin (e.g., direct renin concentration vs plasma renin activity), justify any scaling (ARR = ALD/(REN×10)), and provide references. Use the same system when interpreting PA screening.

Primary outcome, multiplicity, and model specification

Declare a single primary endpoint (e.g., ΔARR at 1 month) and the primary contrast (within‑patient change; or difference‑in‑change DKD vs non‑DKD). Address multiplicity (Holm/FDR) across multiple endpoints/timepoints. Provide full model formulas with covariates and diagnostics (normality of residuals, linearity, collinearity).

Causal wording vs observational design

Reframe as association given the single‑arm pre/post design. Avoid mechanistic claims (e.g., “SGLT‑2i weakens RAAS–glucose coupling”); prefer “correlations attenuated after treatment.”

Sample size justification linkage

Power calculation targets a between‑group renin difference; the main analyses include within‑subject changes and regression models. Align the power target with the declared primary outcome/contrast and cite the source for Δ and SD assumptions.

Handling of non‑normality and dispersion

Outcomes such as UACR display heavy tails (very large SDs). Provide median (IQR) alongside mean (SD); consider robust regression or transformation. State normality checks; default to non‑parametric tests where appropriate.

Terminology consistency

Standardize terms: DKD (not intermixing DN), ARR (not AAR), fasting blood glucose as FBG (avoid GLU in places). Ensure figure/legend labels match text.

Discussion/Conclusions

• Tone down causal attributions; emphasize concurrent changes and potential confounding. Clarify the clinical caution about PA screening in the context of your precise ARR computation/assay and standard cut‑offs. Expand limitations (single center, washout realities, single‑arm design, regression to the mean).

Ethical clarity on concomitant therapies

RAAS endpoints are sensitive to ACEi/ARB/mineralocorticoid antagonists. Clarify whether these were stopped and for how long; provide baseline concomitant medications by subgroup and discuss potential residual effects.

C) Figures and Tables – Technical Assessment

• Tables: In the outcomes table (e.g., Δ from baseline), clarify that asterisks denote within‑group change vs baseline, whereas the p‑value column compares Δ between groups. Specify tests used per row (paired t/Wilcoxon; two‑sample t/Mann–Whitney) and show units (e.g., UACR mg/g). Ensure consistent decimal precision and spacing around ±.

• Figures: Ensure axis units are shown for RAAS markers; define error bars (SD vs SEM vs 95% CI); label DKD vs non‑DKD and timepoints consistently. Clean legends (remove duplicated text, typos) and align terminology across text and graphics. Provide cost‑effectiveness style plots only if relevant; otherwise, keep to biological/clinical outcomes.

D) Line-Referenced Language and Formatting Corrections

Below are concrete edits. Where exact line numbers vary across versions, use the section/phrase to locate the sentence.

Abstract – Results sentence with ARR units

“ARR decreased by −0.59±1.19 ng/L …” → “ARR decreased (mean Δ −0.59±1.19; unitless).” Remove units; ARR is a ratio.

Abstract – Wording

Replace “ditto; p>0.05” with “between‑group difference was not statistically significant (p>0.05).”

Methods – Intervention duration

Replace conflicting statements (“one month” vs “three months”) with the final, consistent schedule: “Participants received dapagliflozin for 3 months; assessments occurred at baseline, 1 month, and 3 months.”

Methods – ARR formula/units

After “ARR = ALD/(REN×10),” specify ALD and REN assays and units, justify the ×10 scaling with a reference, and remove units when reporting ARR values.

Eligibility – Washout windows and examples

Ensure drug classes and washout intervals are consistent (e.g., ACEi/ARB/diuretics/CCB: ≥2 months; MRA/β‑blockers/direct renin inhibitors: ≥4 weeks), and correct “trazodone” to “terazosin” as the α‑blocker example.

Results – Table notes

In the outcomes table, add: “Asterisk denotes within‑group change vs baseline; p‑value compares change between DKD and non‑DKD.” Add units for UACR (mg/g) and maintain consistent decimals.

Discussion – Causal phrasing

Change “SGLT‑2 inhibitors weaken the RAAS–glucose interaction” to “After SGLT‑2 inhibitor initiation, correlations between RAAS markers and glycemic indices were attenuated.”

Limitations – Typos/artifacts

Correct artifacts like “3one‑month,” “threewo time points,” and fused words (e.g., “slowingdelaying”). Standardize to “slowing or delaying.”

Global – Terminology

Use DKD (not DN), ARR (not AAR), FBG (avoid GLU). Ensure consistency in figures/legends/tables and across text.

Global – Formatting

Remove tracked‑changes residues and “Formatted:” comments; standardize spacing around ± and between numbers/units; ensure subject‑verb agreement in Statistics.

**Do you want your identity to be public for this peer review?** For information about this choice, including consent withdrawal, please see our Privacy Policy

Reviewer #2: **Yes: ** Bala Nimmana

Reviewer #4: No

---

## [Author Response · Author response to Decision Letter 2]

27 Sep 2025

2025.09.26

Dear Dr. Zhanjun Jia:

Re: PONE-D-25-15425R1

Title: Effects of SGLT-2 inhibitors on renin-angiotensin-aldosterone system and their correlation with glucose metabolism in type 2 diabetes mellitus patients with hypertension: A prospective study

Thank you for your kind letter of “PLOS ONE PONE-D-25-15425R1 Revision Request” on September 18, 2025. We thank the editorial requests for the corrections, and we deeply apologized for the considerable inconvenience caused to you due to our errors exist. We have now revised the manuscript according to the concerns raised in the reviews. A detailed copy of the authors' “point-by-point response to the reviewer's comments” is attached, and specific changes addressing reviewers' comments are marked.

Again, we would like to thank you for taking the time to review our manuscript. We would be most grateful for your kind consideration of this revised manuscript for potential publication in PLOS ONE.

We look forward to hearing from you at your earliest convenience.

Sincerely yours,

Ningning Wang

Corresponding author:

Baohong Yue, Professor, PhD

Address: Department of Laboratory Medicine, The First Affiliated Hospital of Zhengzhou University, Zhengzhou, P. R. China

Responses to the Reviewer’s comments:

Re: PONE-D-25-15425R1

Title: Effects of SGLT-2 inhibitors on renin-angiotensin-aldosterone system and their correlation with glucose metabolism in type 2 diabetes mellitus patients with hypertension: A prospective study

We would like to acknowledge the reviewer’s insightful comments and valuable suggestions. In addition, we thank the reviewer for helping us improve the present manuscript. We have addressed the concerns raised by the reviewer as detailed below.

Reviewer #2

C1: The extension of follow-up from one month to three months substantially strengthens the study’s validity and addresses earlier concerns regarding short-term variability in RAAS parameters.

A1: We appreciate the favorable comments provided by the reviewer.

C2: Clarifying the sampling method and recruitment period enhances reproducibility and transparency.

A2: We appreciate the favorable comments provided by the reviewer.

C3: The explicit description of inclusion/exclusion criteria and medication washout timelines reduces ambiguity in patient selection.

A3: We appreciate the favorable comments provided by the reviewer.

C4: Inclusion of sample size estimation addresses concerns about statistical power.

A4: We appreciate the favorable comments provided by the reviewer.

C5: Discuss more explicitly the clinical implications of persistent UACR elevation in DKD patients post-treatment.

A5: We appreciate the favorable comments provided by the reviewer.

C6: Consider adding a limitations paragraph addressing residual confounding due to convenience sampling and single-center design.

A6: Great thanks for your comment. We have added a limitation in the Discussion as follows “Finally, this study recruited patients through convenience consecutive sampling at a single center. Despite employing robust statistical techniques, the possibility of residual, unmeasured confounding factors cannot be excluded. Consequently, further prospective multicenter studies with larger sample size are warranted and should include patients with confirmed PA.”

Reviewer #4

A) Executive Summary and Overall Recommendation

C1: Study focus: Prospective, pre–post evaluation of SGLT-2 inhibitor (dapagliflozin) effects on RAAS components (aldosterone, renin, aldosterone–renin ratio [ARR]) and their relationships with glucose metabolism among adults with type 2 diabetes and hypertension, including DKD vs. non-DKD subgroups.

Main findings: Within 1–3 months of treatment, renin increased, ARR decreased, and glycemic indices improved. Authors warn about potential false-negative primary aldosteronism (PA) screening shortly after initiating SGLT-2 inhibitors.

Verdict: Promising and clinically relevant dataset, but requires major revisions to resolve methodological inconsistencies (treatment duration; medication washout), clarify ARR computation/units, specify statistical models and multiplicity control, ensure consistent terminology, and clean language/formatting.

A1: We appreciate the favorable comments provided by the reviewer. We have addressed the concerns raised, as detailed below.

B) Major Scientific and Statistical Issues (Actionable)

C2: Internal inconsistency in intervention period

Abstract and Results imply a 3‑month protocol, while Methods elsewhere state “one month” and then analyze baseline vs 1 and 3 months. Reconcile everywhere (Abstract, Methods–Intervention, Figure captions, Limitations/Conclusions). State the exact dosing schedule and assessment timepoints (baseline, 1 month, 3 months).

A2: Great thanks for your comment. We have removed tracked-change residues and standardized the dosing schedule (3 months) and assessment timepoints (baseline, 1 month, 3 months) consistently throughout the entire text.

C3: ARR units and computation ARR

ARR is unitless. Remove units (e.g., “ARR −0.59±1.19 ng/L” is incorrect). Specify assays and units for aldosterone and renin (e.g., direct renin concentration vs plasma renin activity), justify any scaling (ARR = ALD/(REN×10)), and provide references. Use the same system when interpreting PA screening. ARR

A3: Great thanks for your comment. We apologize for our errors exist, and have removed the incorrect ARR units in the Abstract and Section 3.3. As stated in reference 8, when ALD was measured in ng/dL and REN in ng/L, the common cut-off value for the ARR was around 5.7(refer to the table below). In our study, both REN and ALD were measured in ng/L. Since 1 ng/L equals 10 ng/dL, the ARR was calculated using the formula: ARR = ALD / (REN × 10).

C4: Primary outcome, multiplicity, and model specification

Declare a single primary endpoint (e.g., ΔARR at 1 month) and the primary contrast (within‑patient change; or difference‑in‑change DKD vs non‑DKD). Address multiplicity (Holm/FDR) across multiple endpoints/timepoints. Provide full model formulas with covariates and diagnostics (normality of residuals, linearity, collinearity).

A4: Great thanks for your comment. We have added the primary and secondary outcomes in Abstract and Section 2.3: “The primary outcome measure was the change in renin (REN) levels during outpatient visits at baseline, 1 month, and 3 months. The secondary outcome was the change in other metabolic biomarkers at baseline and 3-month visit.”

The multiplicity across multiple timepoints was adjusted using the Bonferroni test, with the significance level adjusted to 0.017. Therefore, we have revised the significance level indicated by the asterisk in the legends of Figure 3 and Table 3 from “*p < 0.05 vs. baseline” to “*p < 0.017 vs. baseline”. The Abstract and Section 3.3 of the Results have also been appropriately revised. The full models with covariates (Model 1 and Model 2) have already included in Sections 2.4 and 3.5.

C5: Causal wording vs observational design

Reframe as association given the single‑arm pre/post design. Avoid mechanistic claims (e.g., “SGLT‑2i weakens RAAS–glucose coupling”); prefer “correlations attenuated after treatment.”

A5: Great thanks for your comment. We have revised the statement as “The correlations between RAAS markers and glucose metabolism indices were significantly attenuated, potentially due to lowered blood glucose levels after SGLT‑2is treatment” in the Abstract, Discussion and Conclusions.

C6: Sample size justification linkage

Power calculation targets a between‑group renin difference; the main analyses include within‑subject changes and regression models. Align the power target with the declared primary outcome/contrast and cite the source for Δ and SD assumptions.

A6: Great thanks for your comment. The reference [24] for “the source for mean difference and SD assumptions” has been added to Section 2.4 of the Methods.

C7: Handling of non‑normality and dispersion

Outcomes such as UACR display heavy tails (very large SDs). Provide median (IQR) alongside mean (SD); consider robust regression or transformation. State normality checks; default to non‑parametric tests where appropriate.

A7: Great thanks for your comment. Continuous variables that followed non-normal distribution (Diabetic duration, BMI, TG, AII, ALD, REN, ARR, UACR, CP, HOMA-β, HOMA-IR) were presented as median with interquartile range in Tables 1, 3, S1, S2 and S3. These non-normally distributed data were analyzed using non-parametric tests.

Tables S1 and S2 presented the specific biochemical indicator values (including UACR, presented as median with IQR) before and after treatment for both groups. The results of the statistical analysis were consistent with those shown in Table 2.

Moreover, the non-normally distributed variables (AII, ALD, REN, ARR, UACR, CP, HOMA-β, HOMA-IR) were log-transformed to perform multiple linear regression analyses.

C8: Terminology consistency

Standardize terms: DKD (not intermixing DN), ARR (not AAR), fasting blood glucose as FBG (avoid GLU in places). Ensure figure/legend labels match text.

A8: Great thanks for your comment. We have removed all inconsistent terminology and tracked‑change residues from the entire manuscript. The modified parts have been highlighted.

Discussion/Conclusions

C9: Tone down causal attributions; emphasize concurrent changes and potential confounding. Clarify the clinical caution about PA screening in the context of your precise ARR computation/assay and standard cut‑offs. Expand limitations (single center, washout realities, single‑arm design, regression to the mean).

A9: Great thanks for your comment. We have toned down causal attributions; for example, we changed “SGLT‑2i weakens RAAS–glucose coupling” to “correlations were attenuated after treatment.”

We have added a limitation in the Discussion as follows “Finally, this study recruited patients through convenience consecutive sampling at a single center. Despite employing robust statistical techniques, the possibility of residual, unmeasured confounding factors cannot be excluded. Consequently, further prospective multicenter studies with larger sample size are warranted and should include patients with confirmed PA.”

C10: Ethical clarity on concomitant therapies

RAAS endpoints are sensitive to ACEi/ARB/mineralocorticoid antagonists. Clarify whether these were stopped and for how long; provide baseline concomitant medications by subgroup and discuss potential residual effects.

A10: Great thanks for your comment. The inclusion criteria in Section 2.2 of the Methods already specified the concomitant drug conditions as follows“Patients had not taken any agents that could markedly affect the ARR, such as angiotensin converting enzyme inhibitors, angiotensin receptor blockers, diuretics, or calcium channel blockers within the last 2 months, or aldosterone receptor antagonists, β-blockers, or direct renin inhibitors within the last 4 months [8]; For patients with poorly controlled blood pressure (≥160/100 mmHg), medications with less effect on RAAS , such as α-blockers (e.g., terazosin), were administered.”

C) Figures and Tables – Technical Assessment

C11: Tables: In the outcomes table (e.g., Δ from baseline), clarify that asterisks denote within‑group change vs baseline, whereas the p‑value column compares Δ between groups. Specify tests used per row (paired t/Wilcoxon; two‑sample t/Mann–Whitney) and show units (e.g., UACR mg/g). Ensure consistent decimal precision and spacing around ±.

A11: Great thanks for your comment. Explanations for the above questions have been added to the labels in Table 2 as follows “Tables S1 and S2 presented the specific values and the results of the statistical analysis before and after treatment for both groups. Asterisk denotes within‑group change vs. baseline (a, Wilcoxon matched pairs signed rank test; the other rows: paired t-test); the p‑value column compares the difference between DKD and non‑DKD (two‑sample t-test). (*p<0.05 baseline vs. 3 months)” The unit of UACR has also been added.

C12: Figures: Ensure axis units are shown for RAAS markers; define error bars (SD vs SEM vs 95% CI); label DKD vs non‑DKD and timepoints consistently. Clean legends (remove duplicated text, typos) and align terminology across text and graphics. Provide cost‑effectiveness style plots only if relevant; otherwise, keep to biological/clinical outcomes.

A12: Great thanks for your comment. We have defined the error bars in the legend of Fig. 3 as follows: “Data are expressed as median with interquartile range.” We have removed all inconsistent terminology and tracked‑change residues from the legends.

D) Line-Referenced Language and Formatting Corrections

Below are concrete edits. Where exact line numbers vary across versions, use the section/phrase to locate the sentence.

C13: Abstract – Results sentence with ARR units

“ARR decreased by −0.59±1.19 ng/L …” → “ARR decreased (mean Δ −0.59±1.19; unitless).” Remove units; ARR is a ratio.

A13: Great thanks for your comment. We apologize for our errors exist, and have removed the incorrect ARR units.

C14: Abstract – Wording

Replace “ditto; p>0.05” with “between‑group difference was not statistically significant (p>0.05).”

A14: Great thanks for your comment. We have replaced “ditto; p>0.05” with “between‑group difference, p>0.05.” in the Abstract.

C15: Methods – Intervention duration

Replace conflicting statements (“one month” vs “three months”) with the final, consistent schedule: “Participants received dapagliflozin for 3 months; assessments occurred at baseline, 1 month, and 3 months.”

A15: Great thanks for your comment. We have removed tracked-change residues and standardized the dosing schedule (3 months) and assessment timepoints (baseline, 1 month, 3 months) consistently throughout the entire text.

C16: Methods – ARR formula/units

After “ARR = ALD/(REN×10),” specify ALD and REN assays and units, justify the ×10 scaling with a reference, and remove units when reporting ARR values.

A16: Great thanks for your comment. We apologize for our errors exist, and have removed the incorrect ARR units. As stated in reference 8, when ALD was measured in ng/dL and REN in ng/L, the common cut-off value for the ARR was around 5.7(refer to the table below). In our study, both REN and ALD were measured in ng/L. Since 1 ng/L equals 10 ng/dL, the ARR was calculated using the formula: ARR = ALD / (REN ×10).

C17: Eligibility – Washout windows and examples

Ensure drug classes and washout intervals are consistent (e.g., ACEi/ARB/diuretics/CCB: ≥2 months; MRA/β‑blockers/direct renin inhibitors: ≥4 weeks), and correct “trazodone” to “terazosin” as the α‑blocker example.

A17: Great thanks for your comment. We have ensured that drug classes and washout intervals are consistent and corrected “trazodone” to “terazosin” as the α‑blocker example in Section 2.2 of Methods.

C18: Results – Table notes

In the outcomes table, add: “Asterisk denotes within‑group change vs baseline; p‑value compares change between DKD and non‑DKD.” Add units for UACR (mg/g) and maintain consistent decimals.

A18: Great thanks for your comment. Explanations for the above questions have been added to the labels in Table 2 as follows “Tables S1 and S2 presented the specific values and the results of the statistical analysis before and after treatment for both groups. Asterisk denotes within‑group change vs. baseline (a, Wilcoxon matched pairs signed rank test; the other rows: paired t-test); the p‑value column compares the difference between DKD and non‑DKD (two‑sample t-test). (*p<0.05 baseline vs. 3 months)” The unit of UACR has also been added.

C19: Discussion – Causal phrasing

Change “SGLT‑2 inhibitors weaken the RAAS–glucose interaction” to “After SGLT‑2 inhibitor initiation, correlations between RAAS markers and glycemic indices were attenuated.”

A19: Great thanks for your comment. We have revised the statement as “The correlations between RAAS markers and glucose metabolism indices were significantly attenuated, potentially due to lowered blood glucose levels after SGLT‑2is treatment” in the Abstract, Discussion and Conclusions.

C20: Limitati

---

## [Decision Letter · Decision Letter 2]

21 Oct 2025

Effects of SGLT-2 inhibitors on renin-angiotensin-aldosterone system and their correlation with glucose metabolism in type 2 diabetes mellitus patients with hypertension: A prospective study

PONE-D-25-15425R2

Dear Dr. Wang,

We’re pleased to inform you that your manuscript has been judged scientifically suitable for publication and will be formally accepted for publication once it meets all outstanding technical requirements.

Kind regards,

Zhanjun Jia

Academic Editor

PLOS ONE

Additional Editor Comments (optional):

Reviewers' comments:

Reviewer's Responses to Questions

**Comments to the Author**

Reviewer #4: All comments have been addressed

2. Is the manuscript technically sound, and do the data support the conclusions?

Reviewer #4: Yes

3. Has the statistical analysis been performed appropriately and rigorously?

Reviewer #4: Yes

4. Have the authors made all data underlying the findings in their manuscript fully available?

Reviewer #4: Yes

5. Is the manuscript presented in an intelligible fashion and written in standard English?

Reviewer #4: Yes

Reviewer #4: The revised manuscript satisfactorily integrates ≥90 % of my previous recommendations. Only minor editorial cleaning and clarification of model diagnostics and one typo correction are still needed

**Do you want your identity to be public for this peer review?** For information about this choice, including consent withdrawal, please see our Privacy Policy

Reviewer #4: No

---

## [Editor Report · Acceptance letter]

PONE-D-25-15425R2

PLOS ONE

Dear Dr. Wang,

I'm pleased to inform you that your manuscript has been deemed suitable for publication in PLOS ONE. Congratulations! Your manuscript is now being handed over to our production team.

Kind regards,

on behalf of

Dr. Zhanjun Jia

Academic Editor

PLOS ONE